# FEW-SHOT ONE-CLASS CLASSIFICATION VIA META-LEARNING

## ABSTRACT

Although few-shot learning and one-class classification (OCC), i.e. learning a binary classifier with data from only one class, have been separately well studied, their intersection remains rather unexplored. Our work addresses the few-shot OCC problem and presents a meta-learning approach that requires only few data examples from only one class to adapt to unseen tasks. The proposed method builds upon the model-agnostic meta-learning (MAML) algorithm (Finn et al., 2017) and learns a model initialization particularly suited for learning few-shot OCC tasks. This is done by explicitly optimizing for a parameter initialization which only requires a few gradient steps with one-class minibatches to yield a performance increase on class-balanced test data. We provide a theoretical analysis that explains why our approach works in the few-shot OCC scenario, while other meta-learning algorithms, including MAML, fail. Empirical results on six datasets from the image and time-series domains show that our method substantially outperforms both, classical OCC and few-shot classification approaches, and demonstrate the ability to quickly learn unseen tasks from only few normal class samples. Moreover, we successfully learn anomaly detectors for a real-world application on sensor readings recorded during industrial manufacturing of workpieces with a CNC milling machine using a few examples from the normal class.

## 1 INTRODUCTION

The anomaly detection (AD) task (Chandola et al., 2009; Aggarwal, 2015) consists in differentiating between normal and abnormal data samples. AD applications are common in various domains that involve different data types, including medical diagnosis (Prastawa et al., 2004), cybersecurity (Garcia-Teodoro et al., 2009) and quality control in industrial manufacturing (Scime & Beuth, 2018). Due to the rarity of anomalies, the data underlying AD problems exhibits high class-imbalance. Therefore, AD problems are usually formulated as one-class classification (OCC) problems (Moya et al., 1993), where either only a few or no anomalous data samples are available for training the model (Khan & Madden, 2014). While most of the developed approaches (Khan & Madden, 2014) require a substantial amount of normal data to yield good generalization, in many real-world applications, e.g. in industrial manufacturing, only small datasets are available. Data scarcity can have many reasons: data collection itself might be expensive, e.g. in healthcare, or happens only gradually, such as in a cold-start situation. To enable learning from few examples, various viable meta-learning approaches (Lake et al., 2011; Ravi & Larochelle, 2016; Finn et al., 2017) have been developed. However, they rely on having examples from each of the classification task's classes, which prevents their application to OCC tasks. To the best of our knowledge, the few-shot OCC (FS-OCC) problem has only been addressed by Kozerawski & Turk (2018) in the image domain.

Our contribution is threefold: Firstly, we show that classical OCC approaches fail in the few-shot data regime. Secondly, we provide a theoretical analysis showing that classical gradient-based meta-learning algorithms do not yield initializations suitable for OCC tasks and that second-order derivatives are needed to optimize for such initializations. Thirdly, we propose one-class model-agnostic meta-learning (OC-MAML), a data-domain-agnostic algorithm that quickly learns FS-OCC tasks, to serve as a first, simple and strong baseline for future research in the understudied FS-OCC problem.

OC-MAML builds upon model-agnostic meta-learning (MAML) (Finn et al., 2017), which is a meta-learning method that explicitly optimizes for few-shot learning and yields a model initialization

that enables quick adaptation to a new task using only few of its datapoints. Like MAML, OC-MAML yields model parameters that are easily adaptable to unseen tasks. The difference is that the model initialization delivered by OC-MAML is particularly suited for adaptation to OCC tasks and hence requires few examples from only one class of the target task for good adaptation. We provide a theoretical analysis that shows that OC-MAML explicitly optimizes for parameter initializations which yield performance increase on class-balanced test data by taking only a few gradient steps with one-class minibatches. This is done by maximizing the inner product of gradients computed on different minibatches with *different* class-imbalance rates. While recent meta-learning approaches focused on the few-shot learning problem, i.e. learning to learn with few examples, we extend their use to the OCC problem, i.e. learning to learn with examples from only one class.

We empirically validate our theoretical analysis on six datasets from the image and time-series domains, and demonstrate the robustness and maturity of our approach for real-world application by successfully testing it on a real-world dataset of sensor readings recorded during manufacturing of metal workpieces with a CNC milling machine.

## 2 Approach

### 2.1 Problem statement

Our goal is to learn a one-class classification (OCC) task using only a few examples from the normal class. In the following, we first discuss the unique challenges of the few-shot one-class classification (FS-OCC) problem. Subsequently, we formulate the FS-OCC problem as a meta-learning problem.

In order to perform one-class classification, i.e. differentiate between in-class and out-of-class examples, approximating a *generalized* decision boundary for the normal class is necessary. Learning such a class decision boundary in the few-shot regime can be especially challenging for the following reasons. On the one hand, if the model overfits to the few available datapoints, the class decision boundary would be too restrictive, which would prevent generalization to unseen examples. As a result, some normal samples would be predicted as anomalies. On the other hand, if the model overfits to the majority class, e.g. predicting almost everything as normal, the class decision boundary would overgeneralize, and out-of-class (anomalous) examples would not be detected.

In our meta-learning problem formulation, we assume access to data from classification tasks $T_i^{train}$ sampled from a task distribution $p(T)$ related to our target OCC tasks. In the few-shot classification context, $N$-way $K$-shot learning tasks are usually used to test the learning procedure, in our case the model initialization, yielded by the meta-learning algorithm. An $N$-way $K$-shot classification task includes $K$ examples from each of the $N$ classes that are used for learning this task, after which the trained classifier is tested on a disjoint set of data (Vinyals et al., 2016). When the target task is an OCC task, only examples from one class are available for training, which can be viewed as a 1-way $K$-shot classification task. In order to align with the AD problem, the available examples have to belong to the normal (majority) class, which usually has a lower variance than the anomalous (minority) class. This problem formulation is a prototype for a practical use case where an application-specific anomaly detector is needed and only few normal class examples are available.

### 2.2 Model-Agnostic Meta-Learning

Model-agnostic meta-learning (MAML) (Finn et al., 2017) is an optimization-based meta-learning algorithm upon which we build in our present work. MAML learns a model initialization that enables quick adaptation to unseen tasks using only few data samples. For that, MAML trains a model explicitly for few-shot learning on tasks $T_i$ coming from the same task distribution $p(T)$ as the unseen target task $T_{test}$. In order to assess the model's adaptation ability to *unseen* tasks, the available tasks are divided into mutually disjoint task sets: one for meta-training $S^{tr}$, one for meta-validation $S^{val}$ and one for meta-testing $S^{test}$. Each task $T_i$ is divided into two disjoint sets of data, each of which is used for a particular MAML operation: $D^{tr}$ is used for adaptation and $D^{val}$ is used for validation, i.e. evaluating the adaptation. The adaptation procedure of a model $f_\theta$ to a particular task $T_i$ consists in taking one (or more) gradient descent step(s) using *few* datapoints sampled from $D^{tr}$. We also refer to the adaptation updates as *inner loop updates*.

A good measure for the suitability of the initialization parameters $\theta$ for few-shot adaptation to a considered task $T_i$ is the loss $L_{T_i}^{val}(f_{\theta_i'})$, which is computed on the validation set $D^{val}$ using the task-specific adapted model $f_{\theta_i'}$. In order to optimize for few-shot learning, the model parameters $\theta$ are updated by minimizing the aforementioned loss across all meta-training tasks. This update, called the *outer loop update*, can be expressed as:

$$\theta \leftarrow \theta - \beta \nabla_\theta \sum_{T_i \sim p(T)} L_{T_i}^{val}(f_{\theta_i'}), \tag{1}$$

where $\beta$ is the learning rate used for the outer loop. In order to avoid meta-overfitting, i.e. overfitting to the meta-training tasks, model selection can be done via conducting validation episodes using tasks from $S^{val}$ throughout meta-training. At meta-test time, the few-shot adaptation to unseen tasks from $S^{test}$ is evaluated. We note that, in the case of few-shot classification, $K$ datapoints from *each* class are sampled from $D^{tr}$ for the adaptation, during training, validation and testing.

### 2.3 ONE-CLASS MODEL-AGNOSTIC META-LEARNING

#### 2.3.1 ALGORITHM

The primary contribution of our work is to show that second-order gradient-based meta-learning is a viable approach to the underexplored few-shot one-class classification (FS-OCC) problem. We achieve this by adequately modifying the objective of the adaptation step, i.e. the inner loop up-dates, of the MAML algorithm. We choose to build upon gradient-based meta-learning algorithms, because these were shown to be universal learning algorithm approximators (Finn & Levine, 2017), which means that they could approximate a learning algorithm tailored for FS-OCC. As explained in Section 2.2, MAML optimizes explicitly for few-shot adaptation by creating and using auxil-iary tasks that have the same characteristic as the target tasks, in this case tasks that include only few datapoints for training. Analogously, OC-MAML trains explicitly for quick adaptation to OCC tasks by creating OCC auxiliary tasks for meta-training. Concretely, this is done by modifying the class-imbalance rate (CIR) of the inner loop data batches to match the one of the test task. The meta-training procedure of OC-MAML is described in Algorithm 1 in Appendix A.

As described in Section 1, OCC problems are binary classification scenarios where only few or no minority class samples are available. In order to address both of theses cases, we introduce a hyperparameter ($c$) which sets the CIR of the batch sampled for the inner updates. Hereby, $c$ gives the percentage of the samples belonging to the minority (anomalous) class w.r.t. the total number of samples, e.g. setting $c = 0\%$ means only majority class samples are contained in the data batch. We focus on this latter extreme case, where no anomalous samples are available for learning.

The key difference between MAML and OC-MAML is in the sampling operation of the inner loop batch (operation 5 in Algorithm 1 in Appendix A). By reducing the size of the batch used for the adaptation (via the hyperparameter $K$), MAML trains for few-shot adaptation. OC-MAML extends this approach to train for few-shot one-class adaptation by reducing the CIR of the batch used for adaptation (via the hyperparameter $c$). In order to evaluate the performance of the adapted model on both classes, we use a class-balanced validation batch $B'$ for the outer loop updates. This way, we maximize the performance of the model in recognizing both classes after having *seen* examples from only one class during adaptation. Using OCC tasks for adaptation during meta-training favors model initializations that enable a quick adaptation to OCC tasks over those that require class-balanced tasks. From a representation learning standpoint, OC-MAML learns representations that are not only broadly suitable for the data underlying $p(T)$, but also particularly suited for OCC tasks. In Section 2.3.2, we discuss the unique characteristics of the model initializations yielded by OC-MAML and explain why adapting first-order meta-learning algorithms to the OCC scenario does not yield the targeted results.

#### 2.3.2 THEORETICAL ANALYSIS: WHY DOES OC-MAML WORK ?

In this section we give a theoretical explanation of why OC-MAML works and why it is a more suitable approach than MAML for the few-shot one-class classification (FS-OCC) problem. To address the latter problem, we aim to find a model parameter initialization, from which adaptation using few data examples from only *one* class yields a good performance on both classes, i.e. good

generalization to the class-balanced task. We additionally demonstrate that adapting first-order meta-learning algorithms, e.g. First-Order MAML (FOMAML) (Finn et al., 2017) and Reptile (Nichol & Schulman, 2018), to the OCC scenario as done in OC-MAML, does not yield initializations with the desired characteristics, as it is the case for OC-MAML.

$$g_{MAML} = \overline{g}_2 - \alpha \overline{H}_2 \overline{g}_1 - \alpha \overline{H}_1 \overline{g}_2 + O(\alpha^2)$$
$$= \overline{g}_2 - \alpha \frac{\partial(\overline{g}_1 . \overline{g}_2)}{\partial \phi_1} + O(\alpha^2) \tag{2}$$

By using a Taylor series expansion, Nichol & Schulman (2018) approximate the gradient used in the MAML update. For simplicity of exposition, in Equation 2 we give their results for the case where only 2 gradient-based updates are performed, i.e. one adaptation update on a minibatch including $K$ datapoints from $D^{tr}$ and one meta-update on a minibatch including $Q$ datapoints from $D^{val}$. We use the same notation used by Nichol & Schulman (2018), where $\overline{g}_i$ and $\overline{H}_i$ denote the gradient and Hessian computed on the $i^{th}$ minibatch at the initial parameter point $\phi_1$, and $\alpha$ gives the learning rate. Here it is assumed that the same learning rate is used for the adaptation and meta-updates.

In Equation 2 Nichol & Schulman (2018) demonstrate that MAML partially optimizes for increasing the inner product of the gradients computed on different minibatches. In fact, when gradients from different minibatches have a positive inner product, taking a gradient step using one of them yields a performance increase on the other (Nichol & Schulman, 2018). Equation 2 holds also for OC-MAML. However, in OC-MAML the minibatches 1 and 2 have different class-imbalance rates (CIRs), since the first minibatch includes data from only one class and the second minibatch is class-balanced. Hence, it optimizes for increasing the inner product of the gradients computed on different minibatches with *different* CIRs, while MAML does the same but for different minibatches with the *same* CIR, namely $c = 50\%$. Consequently, OC-MAML optimizes for a parameter initialization from which taking one (or few) gradient step(s) with one-class minibatch(es) results in a performance increase on class-balanced data. In contrast, MAML optimizes for a parameter initialization that requires class-balanced minibatches to yield the same effect (Figure 1 in Appendix A). When adapting to OCC tasks, however, only examples from one class are available. We conclude, therefore, that using minibatches with different CIRs for meta-training, as done in OC-MAML, yields parameter initializations that are more suitable for adapting to OCC tasks.

A natural question is whether applying our modification of MAML, i.e. using only data from the normal class for adaptation during meta-training, to other gradient-based meta-learning algorithms would yield the same desired effect. We investigate this for First-Order MAML (FOMAML) (Finn et al., 2017) and Reptile (Nichol & Schulman, 2018). FOMAML is a first-order approximation of MAML, which ignores the second derivative terms. Reptile is also a first-order meta-learning algorithm that learns an initialization that enables fast adaptation to test tasks using only few examples from *each* class. In the following we demonstrate that adapting the FOMAML and Reptile algorithms to the one-class classification scenario, to which we will refer as OC-FOMAML and OC-Reptile, does *not* result in optimizing for an initialization suitable for OCC tasks, as it is the case for OC-MAML. We note that for OC-Reptile, the first $(N-1)$ batches contain examples from only one class and the last $(N^{th})$ batch is class-balanced. The approximated gradients used in the FOMAML and Reptile updates are given by Equations 3 and 4 (Nichol & Schulman, 2018), respectively.

$$g_{FOMAML} = \overline{g}_2 - \alpha \overline{H}_2 \overline{g}_1 + O(\alpha^2) \tag{3}$$

$$g_{Reptile} = \overline{g}_1 + \overline{g}_2 - \alpha \overline{H}_2 \overline{g}_1 + O(\alpha^2) \tag{4}$$

We note that these equations hold also for OC-FOMAML and OC-Reptile. By taking the expectation over minibatch sampling $\mathbb{E}_{\tau,1,2}$ for a meta-training task $\tau$ and two *class-balanced* minibatches, Nichol & Schulman (2018) establish that $\mathbb{E}_{\tau,1,2}[\overline{H}_1 \overline{g}_2] = \mathbb{E}_{\tau,1,2}[\overline{H}_2 \overline{g}_1]$. Averaging the two sides of the latter equation results in the following:

$$\mathbb{E}_{\tau,1,2}[\overline{H}_2 \overline{g}_1] = \frac{1}{2} \mathbb{E}_{\tau,1,2}[\overline{H}_1 \overline{g}_2 + \overline{H}_2 \overline{g}_1] = \frac{1}{2} \mathbb{E}_{\tau,1,2}[\frac{\partial(\overline{g}_1 . \overline{g}_2)}{\partial \phi_1}] \tag{5}$$

Equation 5 shows that, in expectation, FOMAML and Reptile, like MAML, optimize for increasing the inner product of the gradients computed on different minibatches with the *same* CIR. However, when the minibatches 1 and 2 have different CIRs, which is the case for OC-FOMAML and OC-Reptile, $\mathbb{E}_{\tau,1,2}[\overline{H}_1 \overline{g}_2] \neq \mathbb{E}_{\tau,1,2}[\overline{H}_2 \overline{g}_1]$ and therefore $\mathbb{E}_{\tau,1,2}[\overline{H}_2 \overline{g}_1] \neq \frac{1}{2} \mathbb{E}_{\tau,1,2}[\frac{\partial(\overline{g}_1 . \overline{g}_2)}{\partial \phi_1}]$. Hence,

even though, similarly to OC-MAML, OC-FOMAML and OC-Reptile use minibatches with different CIRs for meta-training, contrarily to OC-MAML, they do *not* optimize for increasing the inner product of the gradients computed on different minibatches with *different* CIRs. The second derivative term $\overline{H}_1 \overline{g}_2$ is, thus, necessary to optimize for an initialization from which performance increase on a class-balanced task is yielded by taking few gradient steps using only data from one class.

## 3 RELATED WORKS

Our proposed method addresses the few-shot one-class classification (FS-OCC) problem, i.e. solving binary classification problems using only *few* datapoints from only *one* class. To the best of our knowledge, this problem was only addressed by Kozerawski & Turk (2018), and exclusively in the image data domain. Kozerawski & Turk (2018) train a feed-forward neural network (FFNN) to learn a transformation from feature vectors, extracted by a CNN pre-trained on ILSVRC 2014 (Russakovsky et al., 2015), to SVM decision boundaries. Hereby, the FFNN is trained on ILSVRC 2012. At test time, an SVM boundary is inferred by using one image of one class from the test task which is then used to classify the test examples. This approach is specific to the image domain since it relies on the availability of very large, well annotated datasets and uses data augmentation techniques specific to the image domain, e.g. mirroring. OC-MAML offers a more general approach to FS-OCC since it is data-domain-agnostic. In fact, it does not require a pre-trained feature extraction model, which might not be available for some data domains, e.g. sensor readings.

### 3.1 FEW-SHOT CLASSIFICATION

Recent few-shot classification approaches may be broadly categorized in optimization-based methods (Ravi & Larochelle, 2016; Finn et al., 2017; Nichol & Schulman, 2018) and metric-based methods (Koch, 2015; Vinyals et al., 2016; Snell et al., 2017; Sung et al., 2018). The optimization-based approaches aim to learn an optimization algorithm (Ravi & Larochelle, 2016) and/or a parameter initialization (Finn et al., 2017; Nichol & Schulman, 2018), that is tailored for few-shot learning. Metric-based techniques learn a metric space where samples belonging to the same class are close together, which facilitates few-shot classification (Koch, 2015; Vinyals et al., 2016; Snell et al., 2017; Sung et al., 2018). Rusu et al. (2018) develops a hybrid method that combines the advantages of both categories. Prior meta-learning approaches to few-shot classification addressed the *N*-way *K*-shot classification problem described in Section 2.1, i.e they only consider neatly class-balanced test classification tasks. Optimization-based techniques require these samples to finetune the learned initialization. In the metric-based methods, these samples are necessary to compute class prototypes (Snell et al., 2017), embeddings needed for verification (Koch, 2015) or relation scores (Sung et al., 2018). Our approach, however, requires only samples from one of the test task's classes for learning. Moreover, while the evaluation of the previous approaches in the classification context was limited to the image domain, we additionally validate OC-MAML on datasets from the time-series domain.

### 3.2 ONE-CLASS CLASSIFICATION

Classical OCC approaches rely on SVMs (Schölkopf et al., 2001; Tax & Duin, 2004) to distinguish between normal and abnormal samples. Pal & Foody (2010) show that the classification accuracy of SVMs decreases with an increasing number of input features, particularly when small datasets are available for training. Hybrid approaches combining SVM-based techniques with feature extractors were developed to compress the input samples in lower dimensional representations (Xu et al., 2015; Erfani et al., 2016; Andrews et al., 2016). Fully deep methods that jointly perform the feature extraction step and the OCC step have also been developed (Ruff et al., 2018). Another category of approaches to OCC uses the reconstruction error of antoencoders (Hinton & Salakhutdinov, 2006) trained with only normal class examples as an anomaly score (Hawkins et al., 2002; An & Cho, 2015; Chen et al., 2017). Yet, determining a decision threshold for such an anomaly score requires labeled data from both classes. Further more recent techniques rely on GANs (Goodfellow et al., 2014) to perform OCC (Schlegl et al., 2017; Ravanbakhsh et al., 2017; Sabokrou et al., 2018). The aforementioned hybrid and fully deep approaches require a considerable amount of data from the OCC task to train the typically highly parametrized models to learn features specific to the normal class. By leveraging auxiliary OCC tasks and explicitly optimizing for few-shot learning, OC-MAML learns a representation that can be adapted to unseen OCC task with only few exaples.

## 4 EXPERIMENTAL EVALUATION

The conducted experiments [1] aim to address the following key questions: $(a)$ How does OC-MAML perform compared to classical one-class classification (OCC) approaches in the few-shot (FS) data regime? $(b)$ Does using OCC tasks for meta-training improve the adaptation to such tasks, as it is the case for few-shot tasks (Finn et al., 2017), and do our theoretical findings (Section 2.3.2) about the differences between the MAML and OC-MAML initializations hold in practice? $(c)$ How does OC-MAML compare to the first-order meta-learning algorithms adapted to the OCC scenario, i.e. OC-FOMAML and OC-Reptile (Section 2.3.2)? $(d)$ How does OC-MAML perform in FS-OCC problems from the time-series domain, which is understudied in the few-shot learning literature?

### 4.1 BASELINES AND DATASETS

This section provides information about the baselines and datasets we use in our experimental evaluation. We compare OC-MAML to the classical one-class classification (OCC) approaches One-Class SVM (OC-SVM) (Schölkopf et al., 2001) and Isolation Forest (IF) (Liu et al., 2008) (Question $(a)$), which we fit to the adaptation set of the test task. Here, we apply PCA to reduce the dimensionality of the data, by choosing the minimum number of eigenvectors so that at least $95\%$ of the variance is preserved as done by Erfani et al. (2016). We additionally tune the inverse length scale $\gamma$ by using $10\%$ of the test set, as done by Ruff et al. (2018), which gives OC-SVM a supervised advantage, compared to the other methods. For a fairer comparison to OC-MAML, where these latter methods also benefit from the meta-training and meta-validation tasks, we additionally train them on embeddings inferred by feature extractors learned on these tasks. Here, we train two types of feature extractors on the meta-training tasks: one is trained in a Multi-Task-Learning (MTL) setting and the other trained using the "Finetune" baseline (FB) (Triantafillou et al., 2019). FB is a few-shot classification approach, where one multi-class classifier is trained with all the classes available in all meta-training tasks, after which, an output layer is finetuned with the few available examples of the target task on top of the learned feature extractor. Moreover, we compare OC-MAML to class-balanced meta-learning algorithms, namely MAML, FOMAML and Reptile, as well as first-order meta-learning algorithms adapted to the OCC scenario, i.e. OC-FOMAML and OC-Reptile (Questions $(b)$ and $(c)$). Experimental details are provided in Appendix B.

We evaluate our approach on six datasets, including 3 from the image domain and 3 from the time-series domain. In the image domain we use 2 few-shot learning benchmark datasets, namely Mini-ImageNet (Ravi & Larochelle, 2016) and Omniglot (Lake et al., 2015), and 1 OCC benchmark dataset, the Multi-Task MNIST (MT-MNIST) dataset. To adapt the datasets to the OCC scenario, we create binary classification tasks, where the normal class contains examples from one class of the initial dataset and the anomalous class contains examples from *multiple* other classes. We create 9 different datasets based on MNIST, where the meta-testing task of each dataset consists in differentiating between a certain digit and the others. We use the same ($10^{th}$) task for meta-validation in all datasets. Since most of the time-series datasets for anomaly detection include data from only one domain and only one normal class, adapting them to the meta-learning problem formulation where several different tasks are required is not possible. Therefore, we create two synthetic time-series (STS) datasets, each including 30 synthetically generated time-series that underlie 30 different anomaly detection tasks, to assess the suitability of OC-MAML to time-series data (Question $(d)$). The time-series underlying the datasets are sawtooth waveforms (STS-Sawtooth) and sine functions (STS-Sine). We propose the STS-datasets as benchmark datasets for the few-shot (one-class) classification problem in the time-series domain. Finally, we validate OC-MAML on a real-world anomaly detection dataset of sensor readings recorded during industrial manufacturing using a CNC milling machine. Various consecutive roughing and finishing operations (pockets, edges, holes, surface finish) were performed on ca. 100 aluminium workpieces to record the CNC Milling Machine Data (CNC-MMD). In Appendix C, we give details about all 6 datasets, the task creation procedures adopted to adapt them to the OCC case, as well as the generation of the STS-datasets.

---

[1] Our OC-MAML implementation and experimental evaluation will be made public upon paper acceptance.

Table 1: Test accuracies (in %) computed on the class-balanced test sets of the test tasks of Mini-ImageNet (1), Omniglot (2), MT-MNIST with $T_{test} = T_0$ (3) and STS-Sawtooth (4). One-class adaptation sets ($c = 0\%$) are used, unless otherwise specified.

| Adaptation set size | $K = 2$ | | | | $K = 10$ | | | |
|---|---|---|---|---|---|---|---|---|
| Model \ Dataset | 1 | 2 | 3 | 4 | 1 | 2 | 3 | 4 |
| FB ($c = 50\%$) | 56.2 | 63.1 | 65.7 | 49.8 | 65.5 | 73.3 | 85.7 | 59.6 |
| MTL ($c = 50\%$) | 55.9 | 50 | 64.6 | 50.1 | 63.1 | 50 | 86.6 | 50.9 |
| FB | 50 | 50.6 | 56.5 | 50 | 50 | 51.2 | 50.3 | 50 |
| MTL | 50 | 50 | 49.7 | 50 | 50.2 | 50 | 45.3 | 50 |
| OC-SVM | 50.2 | 50.6 | 51.2 | 50.1 | 51.2 | 50.4 | 53.6 | 50.5 |
| IF | 50 | 50 | 50 | 50 | 50.7 | 50 | 50.9 | 49.9 |
| FB + OCSVM | 50 | 50 | 55.5 | 50.4 | 51.4 | 58 | 86.6 | 58.3 |
| FB + IF | 50 | 50 | 50 | 50 | 50 | 50 | 76.1 | 51.5 |
| MTL + OCSVM | 50 | 50 | 50 | 50 | 50 | 50.1 | 53.8 | 86.9 |
| MTL + IF | 50 | 50 | 50 | 50 | 50 | 55.7 | 84.2 | 64 |
| OC-MAML (ours) | **65.2** | **91.6** | **86.5** | **96.8** | **72.3** | **93.1** | **90** | **96.7** |

## 4.2 RESULTS AND DISCUSSION

Our results of the comparison between OC-MAML and the classical OCC approaches on the 3 image datasets and on the STS-Sawtooth dataset are summarized in Table 1. OC-MAML consistently outperforms all baselines across all datasets and on both adaptation set sizes. While FB and MTL yield relatively good performance when adapting to class-balanced tasks ($c = 50\%$), they completely fail in adapting to OCC tasks. On the MT-MNIST dataset and the STS-Sawtooth dataset, some of the baselines that combine a feature extractor and a shallow model yield high performance, when the adaptation set size is $K = 10$. Our results of the comparison between OC-MAML and the classical few-shot classification approaches on the 3 image datasets and on the STS-Sawtooth dataset are summarized in Table 2. The results on the other 8 MT-MNIST datasets and on the STS-Sine dataset are presented in Appendix D and are consistent with the results in Tables 1 and 2. We observe that OC-MAML consistently outperforms the other meta-learning algorithms with a substantial margin on all datasets and for both adaptation set sizes. This confirms our theoretical findings (Section 2.3.2) that the initializations yielded by class-balanced meta-learning algorithms as well as OC-FOMAML and OC-Reptile are not optimized for adaptation using data from only one class. These latter yield test accuracies close to $50\%$ showing that they overfitted to the normal class (Table 2 (top)).

In an attempt to increase the performance of the other meta-learning algorithms in the OCC scenario, we add a batch normalization (BN) (Ioffe & Szegedy, 2015) layer immediately before the output layer of the network. This BN operation standardizes the latent features using the mean and standard deviation of the $K$ datapoints available for adaptation, which all belong to the normal class. As a result, this layer would output features with mean close to 0 and standard deviation close to 1 for normal class examples. In contrast, anomalous examples would yield features with other statistics, which simplifies their detection. We hypothesize that by enforcing a mapping of the data to a latent space standardized only by examples from the normal class, the anomalies would clearly fall out of the normal-class-distribution, making their detection easier. We note that the BN layer is used during meta-training as well. Hereby, we fix the learnable scaling ($\gamma$) and centering ($\beta$) parameters of the BN layer to 1 and 0, respectively, to prevent it from shifting the standard distribution.

We find that this simple modification increases the performance of the other meta-learning algorithms on all image datasets. However, OC-MAML without BN still yields the highest results, with only one exception. The higher performance increase when a bigger adaptation set is available ($K = 10$) confirms our hypothesis that enforcing a mapping of the data to a latent space standardized only by examples from the normal class makes the detection of the anomalies easier. In fact, using more examples yields more accurate mean and standard deviation measures, which enables a better approximation of the distribution of the normal class, and hence leads to an improved detection of the anomalies. We also tested these algorithms on networks including a trainable BN layer after each convolutional layer. This yielded comparable results to just adding one non-trainable BN layer

Table 2: Test accuracies (in %) computed on the class-balanced test sets of the test tasks of MiniImageNet (1), Omniglot (2), MT-MNIST with $T_{test} = T_0$ (3) and STS-Sawtooth (4). The results are shown for models without BN (top) and with BN (bottom). One-class adaptation sets ($c = 0\%$) are used, unless otherwise specified.

| Adaptation set size | $K = 2$ | | | | $K = 10$ | | | |
|---|---|---|---|---|---|---|---|---|
| Model \ Dataset | 1 | 2 | 3 | 4 | 1 | 2 | 3 | 4 |
| Reptile ($c = 50\%$) | 50.9 | 54.7 | 65.2 | 92.4 | 51.6 | 60.8 | 83.4 | 87.4 |
| FOMAML ($c = 50\%$) | 52.3 | 68.7 | 75.1 | 73.3 | 65.3 | 55.9 | 60.5 | 78 |
| MAML ($c = 50\%$) | 59.7 | 86.9 | 75.3 | 96.4 | 68.5 | 88.5 | 89.9 | 95.6 |
| Reptile | 50.6 | 53.7 | 50 | 93 | 50 | 60.6 | 50 | 50.7 |
| FOMAML | 50.5 | 50 | 50 | 50 | 50.6 | 50 | 50 | 50 |
| MAML | 52.1 | 65.4 | 64.1 | 86.7 | 50 | 50 | 62.6 | 90.8 |
| OC-Reptile | 50 | 50 | 50 | 50 | 50 | 50 | 64.8 | 50 |
| OC-FOMAML | 54.3 | 52.3 | 67.2 | 68.8 | 50.1 | 50.7 | 50 | 51.1 |
| OC-MAML (ours) | **65.2** | **91.6** | **86.5** | **96.8** | **72.3** | **93.1** | 90 | **96.7** |
| Reptile (w. BN) ($c = 50\%$) | 53 | 55.9 | 74.3 | 65.6 | 58 | 84.9 | 92 | 74.4 |
| FOMAML (w. BN) ($c = 50\%$) | 55.4 | 74.9 | 78.8 | 71.7 | 67.4 | 91 | 88.9 | 76 |
| MAML (w. BN) ($c = 50\%$) | 57 | 60.8 | 78 | 50 | 63.7 | 80.2 | 85.2 | 75.1 |
| Reptile (w. BN) | 52.4 | 50 | 56.2 | 81.7 | 59.3 | 78.2 | 85.5 | 72.5 |
| FOMAML (w. BN) | 59 | 79.6 | 64.5 | 80.7 | 65.7 | 90.6 | 81.6 | 75.8 |
| MAML (w. BN) | 50 | 50 | 50 | 78.2 | 50 | 51.8 | 50 | 50 |
| OC-Reptile (w. BN) | 50 | 50 | 50 | 50.9 | 50 | 50 | 77 | 50.2 |
| OC-FOMAML (w. BN) | 50.1 | 70.2 | 70.4 | 77.6 | 69.7 | 87.5 | **92.1** | 77.9 |
| OC-MAML (w. BN) (ours) | 61.9 | 71.9 | 78.5 | 59.5 | 64.8 | 70 | 89.4 | 71.7 |

Table 3: Test F1-scores of OC-MAML on finishing ($F_i$) and roughing ($R_j$) operations of the CNC-MMD dataset, when only $K = 10$ normal examples are available ($c = 0\%$). The $\pm$ shows 95% confidence intervals over 150 tasks sampled from the test operations (not used for meta-training).

| $F_1$ | $F_2$ | $F_3$ | $F_4$ | $R_1$ | $R_2$ |
|---|---|---|---|---|---|
| $80.0 \pm 2.3\%$ | $89.6 \pm 2.1\%$ | $95.9 \pm 1.1\%$ | $93.6 \pm 3.4\%$ | $85.3 \pm 1.4\%$ | $82.6 \pm 1.4\%$ |

before the output layer. Even though some of the meta-learning algorithms and OCC approaches sometimes outperform OC-MAML (Tables 2, 5, 8, 9), they do not consistently yield high performance in learning FS-OCC tasks across several datasets, as it is the case for OC-MAML. We note that this happens only on few MT-MNIST datasets and explain that by the high overlap between the digit classes underlying the meta-training and meta-testing tasks in the MT-MNIST datasets.

The results of OC-MAML experiments on the CNC-MMD dataset are presented in Table 3. We compute F1-scores for evaluation since the test sets are class-imbalanced. OC-MAML consistently achieves high F1-scores across the 6 different milling processes. This high model performance on the minority class, i.e. in detecting anomalous data samples, is reached by using only $K = 10$ non-anomalous data samples ($c = 0\%$). These results show that OC-MAML yielded a parameter initialization suitable for learning OCC tasks in the time-series data domain. Moreover, the high performance reached show the maturity of this method for industrial real-world applications.

## 5 CONCLUSION

This work addressed the novel and challenging problem of few-shot one-class classification (FS-OCC) and introduced OC-MAML, a robust meta-learning approach to FS-OCC problems that learns model parameters which are easily adaptable to unseen tasks using few examples from only one class. We demonstrated the viability of our method on six datasets from the image time-series domains, including a real-world dataset of industrial sensor readings, where it significantly outperformed classical OCC and few-shot classification methods. Future works could investigate an unsupervised approach to FS-OCC, as done by Hsu et al. (2018) in the class-balanced scenario.

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

## A    OC-MAML: ALGORITHM AND PARAMETER INITIALIZATION

In this section we present the pseudo-code of OC-MAML in Algorithm 1 and a diagram visualizing the parameter initializations yielded by MAML and OC-MAML.

---

**Algorithm 1** Few-shot one-class classification with OC-MAML

---

**Require:** $S^{tr}$: Set of meta-training tasks
**Require:** $\alpha, \beta$: Learning rates
**Require:** $K, Q$: Batch size for the inner and outer updates
**Require:** $c$: CIR for the inner-updates
 1: Randomly initialize $\theta$
 2: **while** not done **do**
 3:     Sample batch of tasks $T_i$ from $S^{tr}$ Let $\{D^{tr}, D^{val}\} = T_i$
 4:     **for all** sampled $T_i$ **do**
 5:         Sample $K$ datapoints $B = \{\mathbf{x}^{(l)}, \mathbf{y}^{(l)}\}$ from $D^{tr}$ such that CIR$= c$
 6:         Initialize $\theta_i' = \theta$
 7:         **for** number of adaptation steps **do**
 8:             Compute adaptation loss $L_{T_i}^{tr}(f_{\theta_i'})$ using $B$
 9:             Compute adapted parameters with gradient descent: $\theta_i' = \theta_i' - \alpha \nabla_{\theta_i'} L_{T_i}^{tr}(f_{\theta_i'})$
10:         **end for**
11:         Sample $Q$ datapoints $B' = \{\mathbf{x}'^{(l)}, \mathbf{y}'^{(l)}\}$ from $D^{val}$
12:         Compute outer loop loss $L_{T_i}^{val}(f_{\theta_i'})$ using $B'$
13:     **end for**
14:     Update $\theta$: $\theta \leftarrow \theta - \beta \nabla_\theta \sum_{T_i} L_{T_i}^{val}(f_{\theta_i'})$
15: **end while**
16: **return** meta-learned parameters $\theta$

---

Figure 1 visualizes the adaptation to a binary classification test task $T_s$ from the parameter initializations yielded by OC-MAML and MAML, denoted by $\theta_{OCMAML}$ and $\theta_{MAML}$ respectively. $\theta_{s,CB}^*$ denotes the optimal parameters for $T_s$. Taking a gradient step using a one-class adaptation set $D_{s,OC}$ (gradient direction denoted by $\nabla L_{s,OC}$), yields a performance increase on $T_s$ when starting from the OC-MAML parameter initialization. In contrast, when starting from the parameter initialization reached by MAML a class-balanced adaptation set $D_{s,CB}$ (gradient direction denoted by $\nabla L_{s,CB}$) is required for a performance increase in $T_s$.

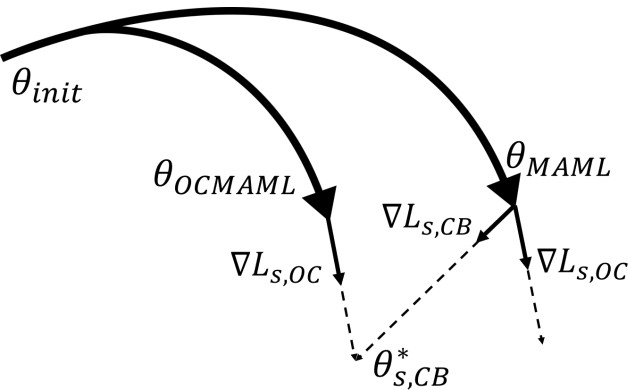

Figure 1: Adaptation to test task $T_s$ from the parameter initializations yielded by OC-MAML and MAML

## B  EXPERIMENT DETAILS

For MT-MNIST, we use the same 4-block convolutional architecture as used by Hsu et al. (2018) for their multi-class MNIST experiments. However, we exclude the batch normalization (Ioffe & Szegedy, 2015) layers, as we want to assess their effect in the OCC case, as discussed in Section 4.2. Each convolutional block includes a 3 x 3 convolutional layer with 32 filters, a 2 x 2 pooling and a ReLU non-linearity. The same model architecture is used for the MiniImageNet experiments as done by Ravi & Larochelle (2016). For the Omniglot experiments, we use the same architecture used by Finn et al. (2017). We also do not include the batch normalization layers for the two latter datasets. On the STS datasets, the model architecture used is composed of 3 modules, each including a 5 x 5 convolutional layer with 32 filters, a 2 x 2 pooling and a ReLU non-linearity. The model architecture used for the CNC-MMD experiments is composed of 4 of these aforementioned modules, except that the convolutional layers in the last two modules include 64 filters. The last layer of all architectures is a linear layer followed by softmax. We note that in the experiments on the time-series datasets (STS and CNC-MMD) 1-D convolutional filters are used.

Table 4: Hyperparameters overview

| Hyperparameter | MT-MNIST | Omniglot | MiniImageNet | STS | CNC-MMD |
|---|---|---|---|---|---|
| Input size | 28 x 28 | 28 x 28 | 84 x 84 x 3 | 128 | 2048 x 3 |
| Meta-learning algorithms (including OC-MAML) | | | | | |
| Outer learning rate ($\beta$) | 0.001 | 0.001 | 0.001 | 0.001 | 0.001 |
| Inner learning rate ($\alpha$) | 0.05 | 0.05 | 0.01 | 0.01 | 0.0001 |
| Task batch size | 8 | 8 | 8 | 8 | 16 |
| Adaptation steps | 5 | 5 | 5 | 10 | 5 |
| Outer loop size ($Q$) | 40 | 20 | 60 | 50 | 4 - 16 |
| MTL and FB | | | | | |
| Batch size | 32 | 32 | 32 | 32 | — |
| Learning rate | 0.05 | 0.05 | 0.01 | 0.01 | — |

Table 4 shows the hyperparameters used in the experiments of each model on the different datasets. We note that we did not fix the outer loop size $Q$ in the experiments on the CNC-MMD dataset, because the sizes and CIRS of the validation sets $D^{val}$ differ across the different tasks. For the meta-learning algorithms, including OC-MAML, we used vanilla SGD in the inner loop and the Adam optimizer (Kingma & Ba, 2014) in the outer loop, as done by Finn et al. (2017). The MTL and FB baselines are also trained with the Adam optimizer.

In the following, we provide details about the meta-training procedure adopted in the meta-learning experiments. We use disjoint sets of data for adaptation ($D^{tr}$) and validation ($D^{val}$) on the meta-training tasks, as it was empirically found to yield better final performance (Nichol & Schulman, 2018). Hereby, the same sets of data are used in the OC-MAML and baseline experiments. In the

MT-MNIST, Omniglot, MiniImageNet and STS experiments, the aforementioned sets of data are class-balanced. The sampling of the batch used for adaptation $B$ ensures that this latter has the appropriate CIR ($c = 50\%$ for MAML, FOMAML and Reptile, and $c = c_{target}$ for OC-MAML, OC-FOMAML and OC-Reptile). For the one-class meta-learning algorithms, $c_{target} = 0\%$, i.e. no anomalous samples of the target task are available, sothat only normal examples are sampled from $D^{tr}$ during meta-training. In order to ensure that class-balanced and one-class meta-learning algorithms are exposed to the same data during meta-training, we move the anomalous examples from the adaptation set of data ($D^{tr}$) to the validation set of data ($D^{val}$). We note that this is only done in the experiments using one-class meta-learning algorithms.

During meta-training, meta-validation episodes are conducted to perform model selection. In order to mimic the adaptation to unseen FS-OCC tasks with CIR $c = c_{target}$ at test time, the CIR of the batches used for adaptation during meta-validation episodes is also set to $c = c_{target}$. We note that the hyperparameter $K$ denotes the total number of datapoints, i.e. batch size, used to perform the adaptation updates, and not the number of datapoints *per class* as done by Finn et al. (2017). Hence, a task with size $K = 10$ and CIR $c = 50\%$ is equivalent to a 2-way 5-shot classification task.

In the following, we provide details about the adaptation to the target task(s) and the subsequent evaluation. In the MT-MNIST and MiniImageNet experiments, we randomly sample 20 adaptation sets from the target task(s)' data, each including $K$ examples with the CIR corresponding to the experiment considered. After each adaptation episode conducted using one of these sets, the adapted model is evaluated on a disjoint class-balanced test set that includes 4,000 images for MT-MNIST and 600 for MiniImageNet. We note that the samples included in the test sets of the test tasks are not used nor for meta-training neither for meta-validation. This results in 20 and 400 (20 adaptation sets created from each of the 20 test classes) different test tasks for MT-MNIST and MiniImageNet, respectively. All the results presented give the mean over all adaptation episodes. Likewise, in the STS experiments, we evaluate the model on 10 different adaptation sets from each of the 5 test tasks. In the CNC-MMD experiments, the 30 tasks created from the target operation are used for adaptation and subsequent evaluation. For each of these target tasks, we randomly sample $K$ datapoints belonging to the normal class that we use for adaptation, and use the rest of the datapoints for testing. We do this 5 times for each target task, which results in 150 testing tasks.

For MTL and FB baselines, as well as all the baseline combining these model with shallow models, i.e. IF and OC-SVM, we use the meta-validation task(s) for model choice, like in the meta-learning experiments. For the MTL baseline, for each validation task, we finetune a fully connected layer on top of the shared multi-task learned layers, as it is done at test time.

## C  DATASETS AND TASK CREATION PROCEDURES

In this Section we provide information about the datasets used and the task creation procedures.

**Multi-task MNIST (MT-MNIST):** We derive 10 binary classification tasks from the MNIST dataset (LeCun et al., 2010), where every task consists in recognizing one of the digits. This is a classical one-class classification benchmark dataset. For a particular task $T_i$, images of the digit $i$ are labeled as normal samples, while out-of-distribution samples, i.e. the other digits, are labeled as anomalous samples. We use 8 tasks for meta-training, 1 for meta-validation and 1 for meta-testing. Hereby, images of digits to be recognized in the validation and test tasks are not used as anomalies in the meta-training tasks. This ensures that the model is not exposed to normal samples from the test task during meta-training. Moreover, the sets of anomalous samples of the meta-training, meta-validation and meta-testing tasks are mutually disjoint. We conduct experiments on 9 MT-MNIST datasets, each of which involves a different target task ($T_0 - T_8$). The task $T_9$ is used as a meta-validation task across all experiments.

**MiniImageNet:** This dataset was proposed by Ravi & Larochelle (2016) and includes 64 classes for training, 16 for validation and 20 for testing, and is a classical challenging benchmark dataset for few-shot learning. To adapt it to the few-shot *one-class* classification setting, we create 64 binary classification tasks for meta-training, each of which consists in differentiating one of the training classes from the others, i.e. the anomalous examples of a task $T_i$ are randomly sampled from the 63

classes with labels different from $i$. We do the same to create 16 meta-validation and 20 meta-testing tasks using the corresponding classes.

**Omniglot:** This dataset was proposed by Lake et al. (2015) and includes 20 instances of 1623 hand-written characters from 50 different alphabets. We generate our meta-training and meta-testing tasks based on the official data split (Lake et al., 2015), where 30 alphabets are reserved for training and 20 for evaluation. For each character class, we create a binary classification task, which consists in differentiating between this character and other characters from the same set (meta-training or meta-testing), i.e. the anomalous examples of a task $T_i$ are randomly sampled from the remaining characters. By removing 80 randomly sampled tasks from the meta-training tasks, we create the meta-validation tasks set.

**Synthetic time-series (STS):** In order to investigate the applicability of OC-MAML to time-series (question $(c)$), we created two datasets, each including 30 synthetically generated time-series that underlie 30 different anomaly detection tasks. The time-series underlying the datasets are sawtooth waveforms (STS-Sawtooth) and sine functions (STS-Sine). Each time-series is generated with random frequencies, amplitudes, noise boundaries, as well as anomaly width and height boundaries. Additionally, the width of the rising ramp as a proportion of the total cycle is sampled randomly for the sawtooth dataset, which results in tasks having rising and falling ramps with different steepness values. The data samples of a particular task are generated by randomly cropping windows of length 128 from the corresponding time-series. We generate 200 normal and 200 anomalous data examples for each task. For each dataset, we randomly choose 20 tasks for meta-training, 5 for meta-validation and 5 for meta-testing. We propose the STS-datasets as benchmark datasets for the few-shot one-class classification problem in the time-series domain, and will make them public upon paper acceptance.

In the following, we give details about the generation procedure adopted to create the STS-Sawtooth dataset. The same steps were conducted to generate the STS-Sine dataset. First, we generate the sawtooth waveforms underlying the different tasks by using the Signal package of the Scipy library (Jones et al., 2001–). Thereafter, a randomly generated noise is applied to each signal. Subsequently, signal segments with window length $l = 128$ are randomly sampled from each noisy signal. These represent the normal, i.e. non-anomalous, examples of the corresponding task. Then, some of the normal examples are randomly chosen, and anomalies are added to them to produce the anomalous examples.

Figure 2: Exemplary normal (left) and anomalous (right) samples belonging to different tasks from the STS-Sawtooth (a and b) and the STS-Sine (c and d) datasets

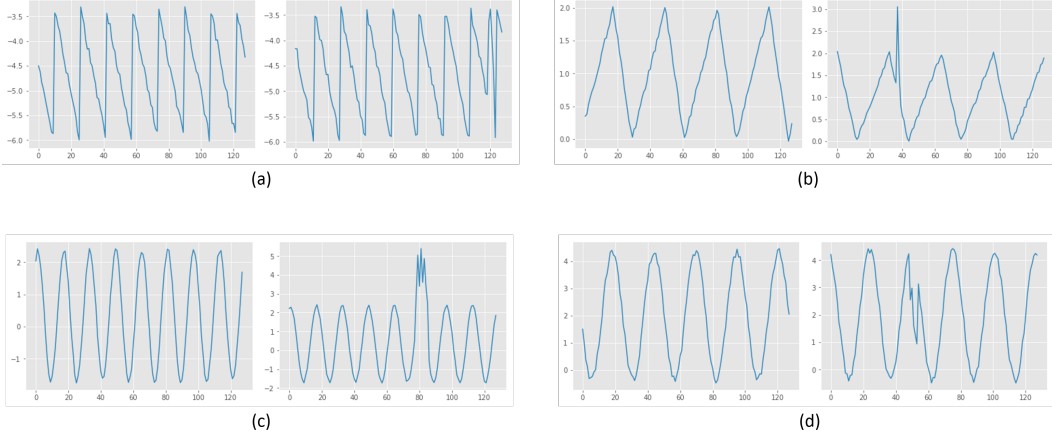

(a)

(b)

(c)

(d)

Figure 2 shows exemplary normal and anomalous samples from the STS-Sawtooth and STS-Sine datasets. In order to increase the variance between the aforementioned synthetic signals underlying the different tasks, we randomly sample the frequency, i.e. the number of periods within the window length $l$, with which each waveform is generated, as well as the amplitude and the vertical position

(see Figure 2). For sawtooth waveforms, we also randomly sample the width of the rising ramp as a proportion of the total cycle between $0\%$ and $100\%$, for each task. Setting this value to $100\%$ and to $0\%$ produces sawtooth waveforms with rising and falling ramps, respectively. Setting it to $50\%$ corresponds to triangle waveforms.

We note that the noise applied to the tasks are randomly sampled from *task-specific* intervals, the boundaries of which are also randomly sampled. Likewise, the width and height of each anomaly is sampled from a random task specific-interval. Moreover, we generate the anomalies of each task, such that half of them have a height between the signal's minimum and maximum (e.g. anomalies $(a)$ and $(d)$ in Figure 2), while the other half can surpass these boundaries, i.e. the anomaly is higher than the normal signal's maximum or lower than its minimum at least at one time step (e.g. anomalies $(b)$ and $(c)$ in Figure 2). We note that an anomalous sample can have more than one anomaly.

We preprocess the data by removing the mean and scaling to unit variance. Hereby, only the available *normal* examples are used for the computation of the mean and the variance. This means that in the experiments, where the target task's size $K = 2$ and only normal samples are available $c = 0\%$, only two examples are used for the mean and variance computation. We note that the time-series in Figure 2 are not preprocessed.

**CNC Milling Machine Data (CNC-MMD):** This dataset consists of ca. 100 aluminum workpieces on which various consecutive roughing and finishing operations (pockets, edges, holes, surface finish) are performed. The sensor readings which were recorded at a rate of 500Hz measure various quantities that are important for the process monitoring including the torques of the various axes. Each run of machining a single workpiece can be seen as a multivariate time-series. We segmented the data of each run in the various operations performed on the workpieces. E.g. one segment would describe the milling of a pocket where another describes a surface finish operation on the workpiece. Since most manufacturing processes are highly efficient, anomalies are quite rare but can be very costly if undetected. For this reason, anomalies were provoked for 6 operations during manufacturing to provide a better basis for the analysis. Anomalies were provoked by creating realistic scenarios for deficient manufacturing. Examples are using a workpiece that exhibits deficiencies which leads to a drop in the torque signal or using rather slightly decalibrated process parameters which induced various irritations to the workpiece surface which harmed production quality. The data was labeled by domain experts from Siemens Digital Industries. It should be noted that this dataset more realistically reflects the data situation in many real application scenarios from industry where anomalies are rare and data is scarce and for this reason training models on huge class-balanced datasets is not an option.

For our experiments, we created 30 tasks per operation by randomly cropping windows of length 2048 from the corresponding time-series of each operation. As a result, the data samples of a particular task $T_i$ cropped from a milling operation $O_j$ correspond to the same trajectory part of $O_j$, but to different workpieces. The task creation procedure ensures that at least two anomalous data samples are available for each task. The resulting tasks include between 15 and 55 normal samples, and between 2 and 4 (9 and 22) anomalous samples for finishing (roughing) operations. We validate our approach on all 6 milling operations in the case where only 10 samples belonging to the normal class ($K = 10$, $c = 0\%$) are available. Given the type of the target milling operation, e.g. finishing, we use the tasks from the other operations of the same type for meta-training. We note that the model is not exposed to any sample belonging to any task of the target operation during training.

We preprocess each of the three signals separately by removing the mean and scaling to unit variance, as done for the STS datasets. Likewise, only the available *normal* examples are used for the computation of the mean and the variance.

Exemplary anomalous signals recorded from a finishing and a roughing operations are shown in Figure 3. These signals are not mean centered and scaled to unit variance. We note that we do not use the labels per time-step, but rather the label "anomalous" is assigned to each time-series that contains at least an anomalous time-step.

Figure 3: Exemplary anomalous samples from a finishing (left) and a roughing (right) operations, where the anomalous time-steps are depicted in red.

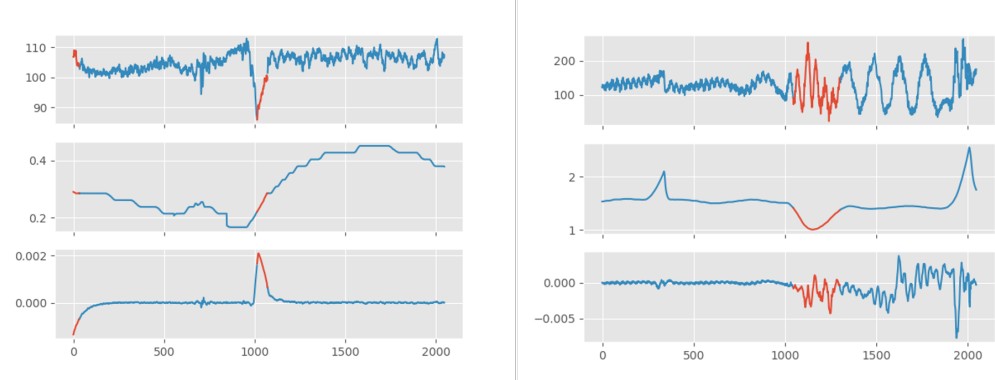

Table 5: Test accuracies (in %) computed on the class-balanced test sets of the test tasks of the MT-MNIST datasets with $T_{test} = T_{1-4}$. One-class adaptation sets ($c = 0\%$) are used, unless otherwise specified.

| Adaptation set size | $K = 2$ | | | | $K = 10$ | | | |
|---|---|---|---|---|---|---|---|---|
| Model \ Dataset | 1 | 2 | 3 | 4 | 1 | 2 | 3 | 4 |
| FB ($c = 50\%$) | 78.8 | 59.8 | 66.7 | 66.8 | 91.9 | 77.3 | 79.9 | 81.5 |
| MTL ($c = 50\%$) | 64.9 | 65 | 59.5 | 56.4 | 91 | 84.6 | 84.4 | 83.3 |
| FB | 53.7 | 56 | 50.7 | 57.1 | 53.6 | 50.7 | 50.2 | 59 |
| MTL | 54 | 46.8 | 41.5 | 52 | 49.4 | 49.6 | 54.7 | 46.1 |
| OC-SVM | 56.9 | 51.5 | 50.5 | 51.8 | 63.7 | 50.2 | 51.2 | 51.5 |
| IF | 50 | 50 | 50 | 50 | 50.9 | 50 | 50.1 | 50 |
| FB + OCSVM | 50.1 | 53.2 | 51.8 | 56.1 | 62.5 | 70.5 | **80.4** | **89.8** |
| FB + IF | 50 | 50 | 50 | 50 | 54.3 | 51.3 | 77.7 | 67.4 |
| MTL + OCSVM | 50 | 50 | 50 | 50 | 50.2 | 52.8 | 54.8 | 50.7 |
| MTL + IF | 50 | 50 | 50 | 50 | 76.5 | 75.5 | 69.3 | 74.4 |
| OC-MAML (ours) | **82.1** | **83.9** | **74.2** | **74.6** | **84.2** | **83.6** | 71.2 | 85.9 |

# D  EXPERIMENTAL RESULTS

In this Section, we present the results of the experiments on the STS-Sine dataset and the 8 further MT-MNIST datasets.

Table 6: Test accuracies (in %) computed on the class-balanced test sets of the test tasks of the MT-MNIST datasets with $T_{test} = T_{5-8}$. One-class adaptation sets ($c = 0\%$) are used, unless otherwise specified.

| Adaptation set size | $K = 2$ | | | | $K = 10$ | | | |
|---|---|---|---|---|---|---|---|---|
| Model \ Dataset | 1 | 2 | 3 | 4 | 1 | 2 | 3 | 4 |
| FB ($c = 50\%$) | 64.6 | 69.8 | 68.9 | 62.9 | 64.4 | 83 | 87.8 | 72.8 |
| MTL ($c = 50\%$) | 60.5 | 71.4 | 65 | 60.6 | 88.4 | 91.4 | 82 | 79.1 |
| FB | 52.2 | 66.5 | 54.3 | 53.8 | 58.3 | 63.5 | 53.6 | 50.1 |
| MTL | 48.5 | 56.2 | 51.1 | 50.1 | 49.9 | 51.4 | 48.5 | 49.6 |
| OC-SVM | 51 | 53.4 | 53.9 | 50.1 | 50.5 | 54 | 54 | 52.2 |
| IF | 50 | 50 | 50 | 50 | 50 | 50.2 | 49.8 | 50.2 |
| FB + OCSVM | 52.2 | 51.2 | 50.5 | 58 | 86.2 | 75 | 84.5 | 80 |
| FB + IF | 50 | 50 | 50 | 50 | 80.4 | 87.2 | 79.2 | 71.4 |
| MTL + OCSVM | 50 | 50 | 50 | 50 | 51 | 59.1 | 71.3 | 75.9 |
| MTL + IF | 50 | 50 | 50 | 50 | 50 | 55.7 | 84.2 | 64 |
| OC-MAML (ours) | **83.3** | **79.5** | **84.6** | **77.3** | **88.4** | **90.7** | **90.8** | **82.6** |

Table 7: Test accuracies (in %) computed on the class-balanced test sets of the test tasks of the STS-Sine dataset. One-class adaptation sets ($c = 0\%$) are used, unless otherwise specified.

| Model \ Adaptation set size | $K = 2$ | $K = 10$ |
|---|---|---|
| FB ($c = 50\%$) | 68.9 | 77.7 |
| MTL ($c = 50\%$) | 64.5 | 91.2 |
| FB | 73.8 | 76.6 |
| MTL | 50 | 50 |
| OC-SVM | 50.2 | 51.3 |
| IF | 50 | 49.9 |
| FB + OCSVM | 52.1 | 65.3 |
| FB + IF | 50 | 62.8 |
| MTL + OCSVM | 50 | 51.9 |
| MTL + IF | 50 | 64.7 |
| OC-MAML (ours) | **99.9** | **99.9** |

Table 8: Test accuracies (in %) computed on the class-balanced test sets of the test tasks of the MT-MNIST datasets with $T_{test} = T_{1-4}$. The results are shown for models without BN (top) and with BN (bottom). One-class adaptation sets ($c = 0\%$) are used, unless otherwise specified.

| Adaptation set size | $K = 2$ | | | | $K = 10$ | | | |
|---|---|---|---|---|---|---|---|---|
| Model \ Dataset | 1 | 2 | 3 | 4 | 1 | 2 | 3 | 4 |
| Reptile ($c = 50\%$) | 69.1 | 71.8 | 52.3 | 51 | 84.4 | 80.4 | 74.2 | 75.2 |
| FOMAML ($c = 50\%$) | 86.7 | 51.5 | 75.1 | 73.3 | 49.8 | 53.9 | 50.3 | 48.2 |
| MAML ($c = 50\%$) | 78.2 | 70.1 | 70 | 53.8 | 90.3 | 86.5 | 81.6 | 84.2 |
| Reptile | 74 | 68.5 | 51.2 | 50 | 50 | 74.4 | 50 | 50 |
| FOMAML | 78.8 | 50 | 50 | 50 | 49.6 | 50 | 50 | 50 |
| MAML | 62.2 | 78.1 | 54.7 | 65.9 | 50 | 83.3 | 56.4 | 50 |
| OC-Reptile | 50 | 50 | 50 | 50 | 50 | 50 | 50 | 50 |
| OC-FOMAML | 60.8 | 49.2 | 50 | 50 | 50 | 50 | 50 | 50 |
| OC-MAML (ours) | 82.1 | **83.9** | **74.2** | **74.6** | 84.2 | 83.6 | 71.2 | **85.9** |
| Reptile (w. BN) ($c = 50\%$) | 63.2 | 60.1 | 58.6 | 57.9 | 92.4 | 87.8 | 84.9 | 81.1 |
| FOMAML (w. BN) ($c = 50\%$) | 80.9 | 68.1 | 59.9 | 61.3 | 93.8 | 86.1 | 82.8 | 80.1 |
| MAML (w. BN) ($c = 50\%$) | 66.9 | 67.1 | 59.1 | 59.6 | 92.3 | 75.1 | 74.1 | 64.5 |
| Reptile (w. BN) | 64.6 | 60.3 | 56.5 | 50 | 61.6 | **84.2** | **82.6** | 83.8 |
| FOMAML (w. BN) | 79.4 | 78.9 | 63.6 | 66.2 | 89 | 83.6 | 62.1 | 76.8 |
| MAML (w. BN) | 50 | 50.1 | 50.2 | 50.3 | 50 | 50.4 | 50 | 50.9 |
| OC-Reptile (w. BN) | 50 | 50 | 50 | 50 | 50 | 50 | 50 | 50 |
| OC-FOMAML (w. BN) | **84.7** | 74.2 | 61.9 | 67.4 | **94.8** | 84 | 81.2 | 79.4 |
| OC-MAML (w. BN) (ours) | 78.4 | 63.7 | 62.4 | 70.8 | 53.3 | 83.3 | 65.7 | 62.2 |

Table 9: Test accuracies (in %) computed on the class-balanced test sets of the test tasks of the MT-MNIST datasets with $T_{test} = T_{5-8}$. The results are shown for models without BN (top) and with BN (bottom). One-class adaptation sets ($c = 0\%$) are used, unless otherwise specified.

| Adaptation set size | $K = 2$ | | | | $K = 10$ | | | |
|---|---|---|---|---|---|---|---|---|
| Model \ Dataset | 1 | 2 | 3 | 4 | 1 | 2 | 3 | 4 |
| Reptile ($c = 50\%$) | 50.4 | 72.4 | 50.1 | 63.7 | 77.6 | 81 | 79.4 | 75.1 |
| FOMAML ($c = 50\%$) | 54.1 | 65.7 | 73.2 | 60.3 | 53.7 | 77.6 | 67.6 | 51.2 |
| MAML ($c = 50\%$) | 76.8 | 72.1 | 75.6 | 66 | 79.9 | 89.2 | 88 | 79.9 |
| Reptile | 50 | 50 | 74.4 | 50 | 50 | 50 | 50 | 50 |
| FOMAML | 50 | 50 | 54 | 50 | 50 | 50 | 62.8 | 50 |
| MAML | 70.7 | 50 | 70.8 | 50 | 50 | 50 | 85.1 | 50 |
| OC-Reptile | 50 | 50 | 50 | 50 | 50 | 50 | 50 | 50 |
| OC-FOMAML | 50.1 | 50 | 50 | 50 | 50 | 50 | 50 | 50 |
| OC-MAML (ours) | **83.3** | **79.5** | **84.6** | **77.3** | **88.4** | **90.7** | 90.8 | 82.6 |
| Reptile (w. BN) ($c = 50\%$) | 59.5 | 62.5 | 62.3 | 59.3 | 84 | 88.6 | 88.4 | 84.2 |
| FOMAML (w. BN) ($c = 50\%$) | 75.3 | 74.4 | 74.1 | 69.8 | 82.6 | 85.2 | 89.8 | 78.9 |
| MAML (w. BN) ($c = 50\%$) | 52.7 | 66.9 | 63.4 | 64.3 | 64.7 | 68.2 | 79.5 | 64.8 |
| Reptile (w. BN) | 51.7 | 68.4 | 73.7 | 50 | 81.8 | 87.6 | 84.6 | 85.3 |
| FOMAML (w. BN) | 70.7 | 78 | 79.7 | 68.7 | 77.5 | 85 | 89.9 | 73.2 |
| MAML (w. BN) | 50 | 51.9 | 50 | 51.9 | 50 | 50 | 50 | 50 |
| OC-Reptile (w. BN) | 50 | 50 | 50 | 50 | 50 | 50 | 50 | 50 |
| OC-FOMAML (w. BN) | 66.7 | 79.1 | 80.3 | 60.7 | 81.1 | 88.8 | **91.8** | **84.7** |
| OC-MAML (w. BN) (ours) | 69.9 | 75.2 | 69.4 | 50.8 | 86.4 | 67.6 | 86.7 | 52.4 |

Table 10: Test accuracies (in %) computed on the class-balanced test sets of the test tasks of the STS-Sine dataset. The results are shown for models without BN (top) and with BN (bottom). One-class adaptation sets ($c = 0\%$) are used, unless otherwise specified.

| Model \ Adaptation set size | $K = 2$ | $K = 10$ |
|---|---|---|
| Reptile ($c = 50\%$) | 99.6 | 99.8 |
| FOMAML ($c = 50\%$) | 91.2 | 89.8 |
| MAML ($c = 50\%$) | 99.5 | 99.4 |
| Reptile | 99.6 | 99.8 |
| FOMAML | 50 | 50.1 |
| MAML | 96.2 | 94.4 |
| OC-Reptile | 50 | 50 |
| OC-FOMAML | 67.1 | 62.7 |
| OC-MAML (ours) | **99.9** | **99.9** |
| Reptile (w. BN) ($c = 50\%$) | 92.2 | 99.2 |
| FOMAML (w. BN) ($c = 50\%$) | 95.8 | 99.4 |
| MAML (w. BN) ($c = 50\%$) | 62.3 | 71.9 |
| Reptile (w. BN) | 60.4 | 76.2 |
| FOMAML (w. BN) | 77.2 | 75 |
| MAML (w. BN) | 61.7 | 87 |
| OC-Reptile (w. BN) | 52.2 | 50.3 |
| OC-FOMAML (w. BN) | 49.8 | 50 |
| OC-MAML (w. BN) (ours) | 50 | 59.9 |

