# OpenReview forum: "Few-Shot One-Class Classification via Meta-Learning"
_ICLR.cc/2020/Conference — Reject_

### Official Review · AnonReviewer2 · 2019-10-23
**Official Blind Review #2**

**Rating:** 1

**Review:**



 This paper tackles an interesting problem, one-class classification or anomaly detection, using a meta-learning approach. The main contribution is to introduce a parameter such that the inner-loop of the meta-learning algorithm better reflects the imbalance which occurs during meta-testing. Results are shown comparing a few simple baselines to both MAML and the modified variant, on a few datasets such as image-based ones (MNIST, miniImageNet), a synthetic dataset, and a real-world time-series example from CNC milling machines.

Overall, the paper presents an interesting problem and awareness that meta-learning might be general enough to solve it well, but provides no real novelty in the approach. The datasets and comparison to other state of art methods (including both other anomaly detection methods and out of distribution methods) is lacking. I suggest the authors perform more rigorous experimentation and focus the paper to be a paper about an understudied problem with rigorous experiments/findings, or improve their method beond the small modification made. Due to these weaknesses, I vote for rejection at this time. Detailed comments are below.

Strengths

  - The problem is interesting and under-studied in the context of deep learning and transferable methods from similar ML problems (e.g. few-shot learning)

  - The method is simple and adapts a state of art in few-shot learning (meta-learning, and specifically MAML)

Weaknesses

  - While I enjoyed reading the paper since it tackles an under-explored problem, it is hard to justify publishing the method/approach at a top machine learning conference. Changing the balance in meta-learning is a relatively obvious modification that one would do to better reflect the problem; I don't think it results in general scientific/ML principles that can be used elsewhere.

  - The relationship to out-of-distribution detection (which some of the experiments, e.g. Multi-task MNIST and miniImagenet essentially test) is not discussed or compared to. How are anomalies defined and is it really different than just being out-of-distribution?

  - The datasets are limited. The MNIST dataset seems to choose a fixed two specific categories for meta-validation and meta-testing, as opposed to doing cross-validation. Results on just one meta-testing seems limited in this case with just one class. In terms of time-series, anomaly detection has been studied for a long time; is there a reason that the authors create a new synthetic dataset? For the milling example, how were anomalies provoked?

  - The baselines do not represent any state of art anomaly detection (e.g. density based, isolation forests, etc.) nor out of distribution detection; the latter especially would likely do extremely well for the simple image examples.

  - There is no analysis of what the difference is in representation (initialization) learning due to the differences between the OCC and FS setup. What are the characteristics of the improved initialization?


One minor comment not reflecting the decision:
  - Exposition: Define the one-class classification problem; it's not common so it would be good to define in the abstract, or mention anomaly detection which is a better-known term.



**Experience Assessment:**

I have published in this field for several years.

**Review Assessment: Checking Correctness Of Derivations And Theory:**

N/A

**Review Assessment: Checking Correctness Of Experiments:**

I carefully checked the experiments.

**Review Assessment: Thoroughness In Paper Reading:**

I read the paper thoroughly.

---

> ### Author Response · Authors · 2019-11-14
> **Our answer regarding weakness 5**
>
> Weakness 5:"There is no analysis of what the difference is in representation (initialization) learning due to the differences between the OCC and FS setup. What are the characteristics of the improved initialization?"
>
>
> We adress the FS-OCC problem in our present work. We assume that your question is about the difference between the initialization yielded by FS-OCC meta-training (OC-MAML) and FS-class-balanced meta-training (MAML). We cover this concern directly in our revised version of the paper (Section 2.3.2):
>
> By analyzing the approximated loss gradients used for the MAML and OC-MAML updates, we come to the following finding: For a given task, OC-MAML optimizes for increasing the inner product of the gradients computed on different minibatches with different class-imbalance rates, namely minibatches containing data from only one class and a class-balanced minibatch (meta-update). If the inner product of the gradients computed on two different minibatches is positive, taking one gradient step using one minibatch leads to an increase in performance on the other minibatch. Consequently, OC-MAML optimizes for a parameter initialization from which taking one (or few) gradient step(s) with minibatch(es) including only normal class data results in a performance increase on class-balanced data. In contrast, MAML optimizes for a parameter initialization that requires class-balanced minibatches to yield the same effect. When adapting to OCC tasks, however, only examples from one class are available. We conclude, therefore, that using minibatches with different class-imbalance rates for meta-training, as done in OC-MAML, yields parameter initializations that are more suitable for adapting to OCC tasks.
>
> We also find that the second-order derivatives are essential to do so, which explains why OC-FOMAML (its first-order approximation) fails to adapt to FS-OCC tasks. Please refer to section 2.3.2 in the revised paper version for a more detailed theoretical analysis of the gradients of the different meta-learning algorithms, in the OCC case.  In the answer to Reviewer 1, we further discuss how batch normalization after the last feature-producing layer can be used to partially counteract this shortcoming. However, despite this modification, the first order meta-learning methods are still outperformed by OC-MAML by a significanat margin.
>
> ----
>
> Minor comment: "Exposition: Define the one-class classification problem; it's not common so it would be good to define in the abstract, or mention anomaly detection which is a better-known term."
>
> Thank you for the comment. We did this in the revised version of the paper.
>
> ----
>
> Thank you again for taking the time to thoroughly read our paper. We noticed that you judged our work with the lowest score. This is really unfortunate since we believe that the topic of few/one-shot one-class classification really deserves a greater attention by the research community due to its wide applicability in many areas where data is naturally scarce and has an extreme class-imbalance. As we address most of your concerns and additional concerns of the other reviewers which clearly improved our paper and our contribution we would really appreciate if you could spare additional time and reevaluate our revised version of the paper and its contributions. Also, if you have further remarks and concerns please feel free to let us know so we can further improve our research contribution.

---

> ### Author Response · Authors · 2019-11-14
> **Our answer regarding weakness 4**
>
> Weakness 4: "The baselines do not represent any state of art anomaly detection (e.g. density based, isolation forests, etc.) nor out of distribution detection; the latter especially would likely do extremely well for the simple image examples."
>
>
> As we mentioned in the paper, we did not compare to the classical one-class classification approaches, since they require high amounts of data and are therefore not applicable in the few-shot regime that we address. It should be noted that OC-SVM and other shallow approaches, e.g. isolation forest (IF) sometimes completely fail in one-class classification even when high amounts of data are available. For example in [3], OC-SVM yields a AUC-ROC of 50% on some of the CIFAR-10 classes, when 5000 datapoints from this class are used for learning.
>
> Upon your request and the request of Reviewer 1, we conducted additional experiments using OC-SVM and IF on the few-shot one-class test tasks. Hereby, we apply PCA to the data where we choose the minimum number of eigenvectors sothat at least 95% of the variance is conserved, as done in [3]. For OC-SVM, we additionally tune the inverse length scale (gamma) by using 10% of the test set, as done in [3]. This gives OC-SVM a supervised advantage, compared to the other baselines.
>
> For a fairer comparison, where these methods also benefit from the data available in the meta-training tasks, we additionally run experiments on the embeddings inferred by the feature extractors of both the "Finetune" baseline (paper 1019 submitted to ICLR 2020) and the Multi-Task-Learning (MTL) baseline. The results are shown in Table 1 in the revised paper version.
>
> As expected, the baselines fail in general to generalize to unseen examples when only few datapoints from the normal class are available. The only exception is the MT-MNIST dataset, where the shallow baselines trained on the extracted embeddings yield good performance  K=10 examples of the normal class are available. We explain this by the fact that in the MT-MNIST dataset, the feature extractor models (MTL and "Finetune") are exposed to most of the anomalies of the test task (8 out of the 9 digit classes present in the test task) during training. Hence, useful embeddings are extracted. We note that, even on the MT-MNIST dataset, OC-MAML still outperfoms all baselines by a significant margin.
>
> We note that in the case where only K=2 examples are available, IF is not applicable, since all trees will have a depth of 1.

---

> ### Author Response · Authors · 2019-11-14
> **Our answer regarding weakness 3**
>
> Weakness 3.1: "The datasets are limited. The MNIST dataset seems to choose a fixed two specific categories for meta-validation and meta-testing, as opposed to doing cross-validation. Results on just one meta-testing seems limited in this case with just one class."
>
>
> In our Multi-Task MNIST (MT-MNIST) experiments, we fixed only the meta-validation task, in which we arbitrarily chose the digit 9 to be the normal class. We actually conductedcross validation , i.e. generated 9 different datasets from MNIST, in each of which one digit (from 0 to 8) is the normal class of the meta-testing task. In the main paper we presented the results of the dataset where the meta-testing task consists in differentiating the digit 0 from the others, but referenced to the Appendix where we presented the results on the other 8 datasets. We note that the results on all datasets were consistent.
>
> During the rebuttal phase we tested our method on a further benchmark dataset for few-shot learning, the Omniglot dataset, where we used the official split of 30 alphabets for meta-training and meta-validation and 20 alphabets for meta-testing. We found consistent results with the other datasets. The results can be seen in the results section of the revised paper version (section 4.2).
>
> In total, we tested our approach on 6 different datasets,  3 image datasets: MT-MNIST (a classical benchmark dataset for One-class classification that actually includes 9 different datasets, one for each class), Omniglot (a classical benchmark dataset for few-shot learning) and MiniImageNet (a challenging benchmark dataset for few-shot learning), as well as 2 synthetic time-series datasets (one based on sine functions and on sawtooth waveforms) and a real-world time-series dataset. As your pointed out that our experimental evaluation is limited, it would be really helpful for us to understand on which datasets we should conduct additional experiments to further strengthen our contribution.
>
> ----
>
> Weakness 3.2: "In terms of time-series, anomaly detection has been studied for a long time; is there a reason that the authors create a new synthetic dataset?"
>
>
> The only reason for this, is that most of the time-series datasets for anomaly detection include data from only one domain and only one normal class, which prevents us from creating different tasks out of these datasets. For the meta-learning problem formulation several different tasks are required.
>
> We generated two synthetic datasets, one based on sine functions and one on sawtooth waveforms, in a way that is suitable for a meta-learning problem. Each of these dataset is composed of 30 different tasks, i.e. 30 different normal signals and anomaly types. Inspired by the impact of the simple toy dataset of sine functions proposed by [1] as a few-shot regression benchmark dataset, we aim for these datasets to be easy-to-use toy datasets for few-shot (one-class) classification on time-series. This latter problem is rather underexplored compared to few-shot classification on image data.
>
> ----
>
> Weakness 3.3:"For the milling example, how were anomalies provoked?"
>
>
> Anomalies were provoked by creating realistic scenarios for deficient manufacturing. Examples are using a workpiece that exhibits deficiencies which leads to a drop in the torque signal or using rather slightly decalibrated process parameters which induced various irritations to the workpiece surface which harmed production quality.

---

> ### Author Response · Authors · 2019-11-14
> **Our answer regarding weaknesses 1 and 2**
>
> In the following we answer the weaknesses you mentioned one by one.
>
>
> Weakness 1: "While I enjoyed reading the paper since it tackles an under-explored problem, it is hard to justify publishing the method/approach at a top machine learning conference. Changing the balance in meta-learning is a relatively obvious modification that one would do to better reflect the problem; I don't think it results in general scientific/ML principles that can be used elsewhere."
>
>
> The modification we made to MAML might be intuitive and simple from an algorithmic point of view. We added a new section to the revised paper version (section 2.3.2) where we provide a theoretical analysis showing that other general meta-learning frameworks, namely FOMAML [1] and Reptile [2], lack the ability to learn one-class classifiers from only few datapoints despite the "simple" modification. In addition, we further back these findings empirically with additional experiments (See Section 4.2 in the revised paper version or our answer for Reviewer 1).
>
> In summary, For a given task, OC-MAML optimizes for a parameter initialization from which taking few gradient steps with one-class minibatches results in a performance increase on class-balanced data. This is done by maximizing the inner product of the gradients of different minibatches with different class-imbalance rates. We refer to section 2.3.2 in the revised version of the paper for more details.
>
> Besides the above mentioned finding, we empirically show that classical OCC methods, such as OC-SVM and isolation forest (IF), completely fail in the low data (few-shot) regime. This demonstrates that the few-shot OCC problem lacks baselines. We therefore believe that our approach will serve as a first, simple and strong benchmark method for future research in this area.
>
>
> ----
>
>
> Weakness 2: "The relationship to out-of-distribution detection (which some of the experiments, e.g. Multi-task MNIST and miniImagenet essentially test) is not discussed or compared to. How are anomalies defined and is it really different than just being out-of-distribution?"
>
>
> In the following we discuss the difference between out-of-distribution detection and one-class classification (or anomaly-detection).
>
> In the out-of-distribution detection literature usually data from a completely different dataset is considered as out-of-distribution, e.g. a model is trained on the CIFAR-10 dataset (of which all data is considered in-distribution) and the test set partially includes data from the TinyImageNet dataset (which would be considered out-of-distribution). In this case, we can say that out-of-dataset-distribution detection is performed. In our experiments we essentially test out-of-class-distribution detection, i.e. the out-of-distribution examples belong to different classes than the normal class but come from the same dataset. Methods from the one-class-classification (and anomaly-detection) literature usually address this latter problem. The out-of-distribution examples (coming from other classes) are in our case are, therefore, closer to the in-distribution examples (coming from the normal class) than in the case of out-of-dataset-distribution case.
>
> In the following we clarify how we define the anomalies and take MiniImageNet (MIN) as an example dataset. MIN reserves 20 classes for testing. In our experiments, a test task include normal examples coming from one of these 20 classes and anomalous examples coming from the other 19 classes. If you consider data belonging to these 19 classes as out-of-distribution, then yes we are performing out-of-distribution detection. However, as mentioned above, this would be out-of-class-distribution detection, i.r. one-class classification.
>
> We address the one-class classification problem and conduct additional experiments to compare to classical OCC approaches (OC-SVM and IF) in the few-shot regime. We do not address the out-of-(dataset)-distribution detection problem and therefore do not compare to the classical methods to solve it.

---

> ### Author Response · Authors · 2019-11-14
> **Summary of our additional contributions**
>
> Thank you for your detailed review and for recognizing that the few-shot one-classification is an under-studied problem and that the simplicity of the proposed method is a strength.
>
>
> We summarize our additional contributions during the rebuttal phase in the following:
>
>
> -Theoretical analysis of why OC-MAML works and why MAML and other first-order meta-learning do not, even when adapted to the OCC case (see Table 1 in the revised paper version).
>
> -Empirical comparison to other gradient-based meta-learning algorithms to validate our theoretical explanation (Table 2 in the revised paper version).
>
> -A modification that increases the performance of class-balanced meta-learning algorithms and first-order one-class meta-learning algorithms, when the test task is a OCC task. However, OC-MAML still yields the highest performance (Table 2 in the revised paper version).
>
> -A comparison to the classical OCC approaches OC-SVM and isolation forest, as well as the "Finetune" baseline (paper 1019 submitted to ICLR 2020), in the few-shot one-class classification scenario (Table 1 in the revised paper version).
>
> -Empirical evaluation of OC-MAML and all the baselines on an additional dataset, the Omniglot dataset, which is a classical benchmark for few-shot learning (see Tables 1 and 2 in the revised paper version).

---

### Official Review · AnonReviewer3 · 2019-10-23
**Official Blind Review #3**

**Rating:** 3

**Review:**

In this paper, the authors have investigated the few shot one classification problem. They have presented a meta-learning approach that requires only few data examples from only one class to adapt to unseen tasks. The proposed method builds upon the model-agnostic meta-learning (MAML) algorithm. I think the topic itself is interesting and I have the following concerns.
(1) The first is about the real requirement of this learning scenario. Although the authors have pointed out some real applications, I think they have been introduced separated. In other words, since this setting is the combination of two previous areas, i.e., one class classification and few-shot learning, I fell that the authors have introduced it by just a combination. What are the unique challenges of this problem? I think these problems should be clarified at first.
(2) The second one is the algorithms itself. Although I have not checked the details, I fell that the authors have prepared this paper in a rough way. The authors have only described the method, without deep analyses answering the question why. For example, the method seems heuristic, without theoretical analysis. In summary, I think this paper likes a technical report, not a research paper.
(3) Although I can catch the main meaning of this paper, it seems that the writing style is not so fluently. I suggested the authors to recognize the presentation.


**Experience Assessment:**

I have published one or two papers in this area.

**Review Assessment: Checking Correctness Of Derivations And Theory:**

I assessed the sensibility of the derivations and theory.

**Review Assessment: Checking Correctness Of Experiments:**

I assessed the sensibility of the experiments.

**Review Assessment: Thoroughness In Paper Reading:**

I made a quick assessment of this paper.

---

> ### Author Response · Authors · 2019-11-14
> **Our answer regarding your other concerns**
>
> Concern 2.1:"The second one is the algorithms itself. Although I have not checked the details, I fell that the authors have prepared this paper in a rough way."
>
>
> We would be grateful for concrete examples of sentences or sections, where you felt that the paper was written in a "rough" way, so that we can improve them.
>
> ----
>
> Concern 2.2:"The authors have only described the method, without deep analyses answering the question why. For example, the method seems heuristic, without theoretical analysis."
>
>
> We added a section (section 2.3.2) in the revised version of the paper, where we give a theoretical explanation of why OC-MAML works. In the following, we briefly summarize our findings. Furthermore, we conducted additional experiments to validate our theoretical analysis empirically. For the results of these experiments please see Table 2 in the revised paper version.
>
> By analyzing the approximated loss gradients used in the MAML and OC-MAML updates, we come to the following finding: For a given task, OC-MAML optimizes for increasing the inner product of the gradients computed on different minibatches with different class-imbalance rates, namely minibatches containing data from only one class and a class-balanced minibatch (meta-update). If the inner product of the gradients computed on two different minibatches is positive, taking one gradient step using one minibatch leads to an increase in performance on the other minibatch. Consequently, OC-MAML optimizes for a parameter initialization from which taking one (or few) gradient step(s) with minibatch(es) including only normal class data results in a performance increase on class-balanced data. In contrast, MAML optimizes for a parameter initialization that requires class-balanced minibatches to yield the same effect. When adapting to OCC tasks, however, only examples from one class are available. We conclude, therefore, that using minibatches with different class-imbalance rates for meta-training, as done in OC-MAML, yields parameter initializations that are more suitable for adapting to OCC tasks.
>
> We also find that the second-order derivative term is essential to do so, which explains why OC-FOMAML (its first-order approximation) fails to adapt to FS-OCC tasks. Please refer to section 2.3.2 of the revised paper version for a more detailed theoretical analysis of the gradients of the different meta-learning algorithms, in the OCC case.
>
> ----
>
> Concern 2.3:"In summary, I think this paper likes a technical report, not a research paper."
>
>
> Our work emerged from a practical and technical situation, where the few-shot one-class regime is common, namely industrial manufacturing. However, we developed a method that is applicable in multiple data domains, i.e. time-series (in particular, sensor data) and images, and therefore can be adopted beyond the initial technical problem. We aim for our approach to be a first and strong baseline for the, as recognized by both other reviewers, relevant and under-studied research problem of few-shot one-class classification in general, i.e. not to the specific case of few-shot anomaly detection on sensor data. We believe that our approach can be of great value to the research community that will explore this challenging and important problem further in the future. We hope that our extensions of the paper give it a stronger research character.
>
> ----
>
> Concern3: "Although I can catch the main meaning of this paper, it seems that the writing style is not so fluently. I suggested the authors to recognize the presentation."
>
>
> As mentioned above, we would be grateful if you could point out some concrete examples of sentences or sections, of which we should improve the writing style. Could you also please elaborate on the last sentence?
>
> Since you mentioned that you have not thoroughly read our paper, we would really appreciate if you could spare the time to reevaluate our revised paper version. Please feel free to raise any further concerns or add some additional comments. As we believe that we have addressed most of your concerns  we would also be very grateful if you could reconsider the rather low scoring for our research contribution.

---

> ### Author Response · Authors · 2019-11-14
> **Our answer regarding your first concern**
>
> In the following we answer your concerns one by one.
>
>
> Concern 1: "The first is about the real requirement of this learning scenario. Although the authors have pointed out some real applications, I think they have been introduced separated. In other words, since this setting is the combination of two previous areas, i.e., one class classification and few-shot learning, I fell that the authors have introduced it by just a combination. What are the unique challenges of this problem? I think these problems should be clarified at first."
>
>
> In section 2.1 "Problem statement" we defined the few-shot one-class classification (FS-OCC) problem and mentioned its challenges. In the following we further clarify the requirements and challenges of the FS-OCC problem. We begin by explaining the requirements and challenges of the OCC problem and the few-shot class-balanced binary classification (FS-CBBC) problem separately.
>
> In the OCC problem, data examples from only one class (the normal class) are available for training a model that has to differentiate between two classes, namely the normal class and the abnormal class. For that a learning algorithm is required,  that enables the trained model to approximate a sufficiently generalized decision boundary for the normal class without overfitting to this class, i.e. approximating a too big decision boundary which results in predicting (almost) everything as normal. Classical OCC approaches, e.g. OC-SVM, usually require high amounts of data from the normal class to be able to learn such a decision boundary. We note that in the OCC scenario there are no restrictions with regards to the amount of data available from the normal class, i.e. access to high amounts of normal data examples is assumed.
>
> In the FS-CBBC problem, few data examples from each of the two classes are available for training a binary classification model. For that a learning algorithm is required, that enables the trained model to approximate a sufficiently generalized decision boundary for the each class without overfitting to the few examples available, i.e. approximating a too tight decision boundary which prevents generalization  to unseen examples from each class. Most of the few-shot classification approaches, e.g. MAML, require access to data examples from each class to yield such generalization with only few examples. We note that in the FS-CBBC scenario there are restrictions with regards to the class distribution of the available training data, namely access to data examples from each of the two classes is assumed.
>
> In the FS-OCC problem, which we address in this work, only few data examples from only one class (the normal class) are available for training a binary classification model. In this scenario restrictions on the data amount (only few examples are available) and on the class distribution in the data (data from only one class is available) are imposed. To address this problem, a learning algorithm is required, that enables the trained model to approximate a sufficiently generalized decision boundary for the normal class using only few of its examples. The unique challenge of this scenario arises from the combination of the challenges of the OCC and the FS-CBBC problems, namely overfitting to the normal class, i.e. predicting (almost) everything as normal, and overfitting to the few available examples, i.e. not being able to generalize to unseen examples.
>
> We updated the section 2.1 "Problem statement" in the revised version of the paper to include more details and clarifications about the requirements and unique challenges of the FS-OCC problem. We would appreciate, if you could give us some feedback on the detailed problem statement given above.

---

> ### Author Response · Authors · 2019-11-14
> **Summary of our additional contributions**
>
> Thank you for your review.
>
>
> We summarize our additional contributions during the rebuttal phase in the following:
>
>
> -Theoretical analysis of why OC-MAML works and why MAML and other first-order meta-learning do not, even when adapted to the OCC case (see Table 1 in the revised paper version).
>
> -Empirical comparison to other gradient-based meta-learning algorithms to validate our theoretical explanation (Table 2 in the revised paper version).
>
> -A modification that increases the performance of class-balanced meta-learning algorithms and first-order one-class meta-learning algorithms, when the test task is a OCC task. However, OC-MAML still yields the highest performance (Table 2 in the revised paper version).
>
> -A comparison to the classical OCC approaches OC-SVM and isolation forest, as well as the "Finetune" baseline (paper 1019 submitted to ICLR 2020), in the few-shot one-class classification scenario (Table 1 in the revised paper version).
>
> -Empirical evaluation of OC-MAML and all the baselines on an additional dataset, the Omniglot dataset, which is a classical benchmark for few-shot learning (see Tables 1 and 2 in the revised paper version).

---

### Official Review · AnonReviewer1 · 2019-10-25
**Official Blind Review #1**

**Rating:** 3

**Review:**

One of promising approach to tackle the few-shot problems is to use meta-learning so that the learner can quickly generalize to an unseen task. One-class classification requires only a set of positive examples to discriminate negative examples from positive examples. The current paper addresses a method of meta-training one-class classifiers in the MAML framework when only a handful of positive examples are available.

---Strength---
- Few-shot one-class classification is a timely subject, which has not be studied yet.
- Meta-training one-class classifiers in the MAML framework seems to be sound.

---Weakness---
- MAML is quite a general meta-training framework, which can be used when parameterized base-learners are updated using gradient methods.  Thus, when parameterized models for one-class classification are used, it is rather easy to meta-train one-class classifiers in the MAML framework.
- Regarding episodic training, in contrast to few-shot classification problems, support sets in episodes have similar positive examples. Thus, fine-tuning baseline method could work well, even without using MAML. Please compare it with the fine-tuning method.

---Comments---
- I assume that the query set in each episode include negative examples, while support sets have only positive examples. Right? What is the value of c (class imbalance rate) in the query set?
- Wouldn't it be  better to focus on experiments with c=0% since one-class classification requires the training with only positive examples?
- What was the baseline one-class classifier? One-class SVM?
- It was mentioned that CLEAR was an earlier work. Then, the empirical comparison with CLEAR should be included when image data is considered.

**Experience Assessment:**

I have published one or two papers in this area.

**Review Assessment: Checking Correctness Of Derivations And Theory:**

I carefully checked the derivations and theory.

**Review Assessment: Checking Correctness Of Experiments:**

I assessed the sensibility of the experiments.

**Review Assessment: Thoroughness In Paper Reading:**

I read the paper at least twice and used my best judgement in assessing the paper.

---

> ### Author Response · Authors · 2019-11-14
> **Our answer to your comments**
>
> Comment 1: "I assume that the query set in each episode include negative examples, while support sets have only positive examples. Right? What is the value of c (class imbalance rate) in the query set?"
>
>
> Yes, support sets include only positive examples, i.e. belonging to the majority (non anomalous) class. Query sets are class-balanced (c=50%) in order to evaluate the model on both classes in the outer loop after adaptation, i.e. the inner loop updates. This way, we optimize the performance of the model on both classes equally after adaptation using only one class.
>
> ----
>
> Comment 2: "Wouldn't it be better to focus on experiments with c=0% since one-class classification requires the training with only positive examples?"
>
>
> We showed the results of c=50% in order to give the reader some reference numbers from the class-balanced case. For example, these results answer the question "How much accuracy gain would having data from both classes yield". As for the experiments with c=1%, we conducted them to show that our approach is not only applicable in the extreme case of one-class classification, but also in the general class-imabalance case. We will focus more on the c=0% case in the revised version of the paper.
>
> ----
>
> Comment 3:"What was the baseline one-class classifier? One-class SVM?"
>
>
> As we mentioned in the paper, we did not compare to the classical one-class classification approaches, since they require high amounts of data and are therefore not applicable in the few-shot regime that we address. It should be noted that OC-SVM and other shallow approaches sometimes completely fail in one-class classification even when high amounts of data are available. For example in [3], OC-SVM yields a AUC-ROC of 50% on some of the CIFAR-10 classes, when 5000 datapoints from this class are used for learning.
>
> Upon your request and the request of Reviewer 2, we conducted additional experiments using the classical OCC approaches OC-SVM and Isolation Forest (IF) on the few-shot one-class test tasks. Hereby, we apply PCA to the data where we choose the minimum number of eigenvectors sothat at least 95% of the variance is conserved, as done in [3]. For OC-SVM, we additionally tune the inverse length scale (gamma) by using 10% of the test set, as done in [3]. This gives OC-SVM a supervised advantage, compared to the other baselines.
>
> For a fairer comparison, where these methods also benefit from the data available in the meta-training tasks, we additionally conducted experiments on the embeddings inferred by the feature extractors of both the "Finetune" baseline and the Multi-Task-Learning (MTL) baseline. The results can be seen in Table 1 of the the revised paper version.
>
> As expected, the baselines fail to generalize to unseen examples when only few datapoints from the normal class are available. The only exception is the MT-MNIST dataset, where the shallow baselines trained on the extracted embeddings yield good performance K=10 examples of the normal class are available. We explain this by the fact that in the MT-MNIST dataset, the feature extractor models (MTL and "Finetune") are exposed to most of the anomalies of the test task (8 out of the 9 digit classes present in the test task) during training. Hence, useful embeddings are extracted. We note that, even on the MT-MNIST dataset, OC-MAML still outperfoms all baselines by a significant margin.
>
> ----
>
> Comment 4:"It was mentioned that CLEAR was an earlier work. Then, the empirical comparison with CLEAR should be included when image data is considered."
>
>
> CLEAR [4] uses a feature extractor trained on ImageNet. Comparing to it on the MiniImageNet dataset would not be fair, as the feature extractor was trained on the test classes. The other datasets that we tested our approach on, MNIST and Omniglot, are composed of grey-scale images. We will not be able to run experiments on datasets that were tested in the CLEAR due to the short rebuttal time. We would like, however, to point out that OC-MAML is data-type-agnostic and was successfully validated on time-series data, to which CLEAR is not applicable.
>
> As we have addressed most of your concerns which clearly improved the quality of the paper, we would really appreciate if you could spare the time to reevaluate our revised paper and adapt the current score accordingly. Please feel free to further comment our additions and raise further concerns.

---

> ### Author Response · Authors · 2019-11-14
> **Our answer regarding weakness 2**
>
> Weakness 2: "Regarding episodic training, in contrast to few-shot classification problems, support sets in episodes have similar positive examples. Thus, fine-tuning baseline method could work well, even without using MAML. Please compare it with the fine-tuning method."
>
>
> We are not sure, we understood the concern in the first sentence correctly. We clarify the episodic training of OC-MAML in the following. In each episode, a new support set is randomly sampled from the majority class data for each meta-training task that was sampled for this episode. Therefore, like in the few-shot class-balanced classification case, support sets differ from episode to episode. The only difference is that in the latter case the support sets include datapoints from all classes. We note that each meta-training task has a different normal class, which means that the support sets of the different tasks have different normal (non-anomalous) examples. Please elaborate on the first sentence if our clarification did not answer your concern.
>
> As requested we compared to the "Finetune" method proposed in the concurrently submitted paper to ICLR 2020 "Meta-Dataset: A Dataset of Datasets for Learning to Learn from Few Examples" (paper 1019). The results are shown in the table 1 in the revised paper version, where we also compare to shallow OCC approaches.
>
> As expected, the "Finetune" baseline performs well in adapting to few-shot class-balanced tasks, but it overfits to the majority class, when finetuned with examples belonging to only one class. As a result, it yields a test accuracy close to 50% on all datasets, like a predictor that always predicts just one outcome, in the one-class classification case.

---

> ### Author Response · Authors · 2019-11-14
> **Our answer regarding weakness 1**
>
> In the following we answer your concerns one by one.
>
> Weakness 1: "MAML is quite a general meta-training framework, which can be used when parameterized base-learners are updated using gradient methods. Thus, when parameterized models for one-class classification are used, it is rather easy to meta-train one-class classifiers in the MAML framework"
>
>
> We adress this concern by providing a theoretical analysis (section 2.3.2 in the revised paper version) and empirically demonstrating that other general meta-learning frameworks, namely FOMAML [1] and Reptile [2], fail to learn one-class classifiers from only few datapoints. For that, we adapt FOMAML and Reptile to the one-class classification scenario by using one-class classifiers for meta-training, i.e. we use examples from only one class for adaptation (in the inner loop) and class-balanced data for the outer loop, as it was done for OC-MAML. We will refer to these algorithms as OC-FOMAML and OC-Reptile, respectively. We note that for OC-Reptile, the first (N-1) batches contain only examples from only one class and the last (Nth) batch is class-balanced. We additionally compare to the class-balanced version of these meta-learning algorithms,  where only class-balanced (CB) batches are used during meta-training (CB-FOMAML and CB-Reptile). The results on the two already used image datasets (Multi-Task MNIST and MiniImageNet) as well as on the during the rebuttal phase added Omniglot dataset are consistent. The results on all datasets can be seen in Table 2 of the revised paper version.
>
> We find that OC-MAML substantially outperforms the other meta-learning algorithms, by a substantial margin on all datasets. As shown in the new section of our revised paper version (section 2.3.2),  OC-MAML is the only meta-learning algorithm that, for a given task,  optimizes for increasing the inner product of the gradients computed on different minibatches with different class-imbalance rates, namely minibatches containing data from only one class and a class-balanced minibatch (meta-update). If the inner product of the gradients computed on two different minibatches is positive, taking one gradient step using one minibatch leads to an increase in performance on the other minibatch. Hence, OC-MAML optimizes for an initialization from which taking a a few gradient steps using a minibatch including datapoints from only one class results in an increased performance on the class-balanced task, i.e. higher performance on both classes. In our analysis of the approximated gradients of the different meta-learning algorithms, we find that the second derivative term is essential to do so.
>
> In an attempt to make the other meta-learning algorithms work in the few-shot one-class scenario, we add a batch normalization (BN) layer immediately before the output layer of the network. This BN layer standardizes the latent features using the mean and std. deviation of the few datapoints available for finetuning, which all belong to the normal class. As a result, this layer would output features with mean close to 0 and std. deviation close to 1 for normal class examples. Anomalous examples would yield features with other statistics, which simplifies their detection. We hypothesize that by enforcing a mapping of the data to a latent space standardized only by examples from the normal class, the detection of the anomalies would be easier, as these clearly fall out of distribution.
>
> The added BN layer is of course used during meta-training as well. Hereby, we do not train the BN parameters of this layer, i.e. the scaling parameter (gamma) is fixed to 1 and the centring parameter (beta) is fixed to 0. We do so, to make sure that the network does not shift the standard distribution. The results are displayed in the Table 2 of the revised paper version.
>
> We find that this simple modification substantially increases the performance of the other meta-learning algorithms, i.e. MAML, FOMAML, Reptile, OC-FOMAML and OC-Reptile, on all image datasets. However, OC-MAML without batch normalization still yields better results. We observe a higher increase in performance when K=10 than when only K=2 examples are available. This confirms our hypothesis that enforcing a mapping of the data to a latent space standardized only by examples from the normal class makes the detection of the anomalies easier. In fact, using more examples yields more accurate mean and std. Deviation measures, which enables a better approximation of the distribution of the normal class and therefore leads to an improved detection of the anomalies.
>
> We also tested these algorithms on networks including a trainable batch normalization layer after each convolutional layer. Comparable results to just adding one non-trainable batch normalization layer before the output layer were yielded.

---

> ### Author Response · Authors · 2019-11-14
> **Summary of our additional contributions**
>
> Thank you for your review and for recognizing the relevance of the few-shot one-classification problem as well as the suitability of meta-learning as an approach to tackle it.
>
>
> We summarize our additional contributions during the rebuttal phase in the following:
>
> -Theoretical analysis of why OC-MAML works and why MAML and other first-order meta-learning do not, even when adapted to the OCC case.
>
> -Empirical comparison to other gradient-based meta-learning algorithms to validate our theoretical explanation.
>
> -A modification that increases the performance of class-balanced meta-learning algorithms and first-order one-class meta-learning algorithms, when the test task is a OCC task. However, OC-MAML still yields the highest performance.
>
> -A comparison to the classical OCC approaches OC-SVM and isolation forest, as well as the "Finetune" baseline (paper 1019 submitted to ICLR 2020), in the few-shot one-class classification scenario.
>
> -Empirical evaluation of OC-MAML and all the baselines on an additional dataset, the Omniglot dataset, which is a classical benchmark for few-shot learning (see results in the revised paper version).

---

### Author Response · Authors · 2019-11-14
**References**

[1]: Chelsea Finn, Pieter Abbeel, and Sergey Levine. Model-agnostic meta-learning for fast adaptation of deep networks. In Proceedings of the 34th International Conference on Machine Learning-Volume 70, pp. 1126–1135. JMLR. org, 2017

[2]: Alex Nichol and John Schulman. Reptile: a scalable metalearning algorithm. arXiv preprint arXiv:1803.02999, 2018

[3]: Lukas Ruff, Robert Vandermeulen, Nico Goernitz, Lucas Deecke, Shoaib Ahmed Siddiqui, Alexander Binder, Emmanuel Müller, and Marius Kloft. Deep one-class classification. In International Conference on Machine Learning, pp. 4393–4402, 2018

[4]: Jedrzej Kozerawski and Matthew Turk. Clear: Cumulative learning for one-shot one-class image recognition. In Proceedings of the IEEE Conference on Computer Vision and Pattern Recognition,pp. 3446–3455, 2018.

---

### Decision · Program_Chairs · 2019-12-19

**Decision:**

Reject

**Comment:**

The authors present a combination of few-shot learning with one-class classification model of problems. The authors use the existing MAML algorithm and build upon it to present a learning algorithm for the problem. As pointed out by the reviewers, the technical contributions of the paper are quite minimal and after the author response period the reviewers have not changed their minds. However, the authors have significantly changed the paper from its initial submission and as of now it needs to be reviewed again. I recommend authors to resubmit their paper to another conference. As of now, I recommend rejection.